

# How emissions uncertainty influences the distribution and radiative impacts of smoke from fires in North America

Therese S. Carter[1], Colette L. Heald[1,2], Jose L. Jimenez[3], Pedro Campuzano-Jost[3], Yutaka Kondo[4], Nobuhiro Moteki[5], Joshua P. Schwarz[6], Christine Wiedinmyer[7], Anton S. Darmenov[8], Arlindo M. da Silva[8], and Johannes W. Kaiser[9]

[1] Civil and Environmental Engineering Department, Massachusetts Institute of Technology, Cambridge, MA 02139, USA
[2] Earth, Atmospheric and Planetary Sciences, Massachusetts Institute of Technology, Cambridge, MA 02139, USA
[3] Cooperative Institute for Research in Environmental Sciences and Department of Chemistry, University of Colorado, Boulder, Colorado 80309, USA
[4] Research Center for Advanced Science and Technology, University of Tokyo, Tokyo, Japan
[5] Department of Earth and Planetary Science, Graduate School of Science, The University of Tokyo, Japan
[6] Chemical Sciences Division, Earth System Research Laboratory, National Oceanic and Atmospheric Administration, Boulder, CO 80305, USA
[7] National Center for Atmospheric Research, Boulder, CO 80307, USA
[8] NASA Goddard Space Flight Center, Greenbelt, MD 20771, USA
[9] Deutscher Wetterdienst, Offenbach am Main, Germany

*Correspondence to*: Therese Carter (tscarter@mit.edu) and Colette Heald (heald@mit.edu)

**Abstract.** Fires and the aerosols that they emit impact air quality, health, and climate, but the abundance and properties of carbonaceous aerosol (both black carbon and organic carbon) from biomass burning (BB) remain uncertain and poorly constrained. We aim to quantify the uncertainties associated with fire emissions and their air quality and radiative impacts from underlying dry matter consumed and emissions factors. To explore this, we compare model simulations from a global chemical transport model, GEOS-Chem, driven by a variety of fire emission inventories with surface and airborne observations of black carbon (BC) and organic aerosol (OA) concentrations and satellite-derived aerosol optical depth (AOD). We focus on two fire detection/burned area-based (FD/BA) inventories using burned area and active fire counts, respectively: the Global Fire Emissions Database version 4 (GFED4s) with small fires and the Fire INventory from NCAR version 1.5 (FINN1.5) and two fire radiative power (FRP)-based approaches: the Quick Fire Emission Dataset version 2.4 (QFED2.4) and the Global Fire Assimilation System version 1.2 (GFAS1.2). We show that, across the inventories, emissions of BB aerosol (BBA) differ by a factor of 4 to 7 over North America and that dry matter differences, not emissions factors, drive this spread. We find that simulations driven by QFED2.4 generally overestimate BC and, to a lesser extent, OA concentrations observations from two fire-influenced aircraft campaigns in North America (ARCTAS and DC3) and from the Interagency Monitoring of Protected Visual Environments (IMPROVE) network, while simulations driven by FINN1.5 substantially underestimate concentrations. The GFED4s and GFAS1.2-driven simulations provide the best agreement with OA and BC mass concentrations at the surface (IMPROVE), BC observed aloft (DC3 and ARCTAS), and AOD observed by MODIS over North America. We also show that a sensitivity simulation including an enhanced source of secondary organic aerosol (SOA) from fires based on the NOAA Fire Lab 2016 experiments produces substantial additional OA; however, the



spread in the primary emissions estimates implies that this magnitude of SOA cannot be either confirmed or ruled out when
comparing the simulations against the observations explored here. Given the substantial uncertainty in fire emissions, as
represented by these four emission inventories, we find a sizeable range in BBA population-weighted exposure over Canada
and the contiguous United States (0.5 to 1.6 µg m$^{-3}$). We also show that the range in the estimated global direct radiative
effect of carbonaceous aerosol from fires (-0.11 to -0.048 W m$^{-2}$) is large and comparable to the direct radiative forcing of
OA (-0.09 W m$^{-2}$) estimated in AR5. Our analysis suggests that fire emissions uncertainty challenges our ability to
accurately characterize the impact of smoke on air quality and climate.
**1 Introduction**
Biomass burning (BB), which includes wildfires in addition to agricultural and other prescribed burning, emits a variety of
trace gases and aerosols, including carbon dioxide, oxides of nitrogen, volatile organic compounds (VOCs), and particulate
matter (PM) (Akagi et al. 2011) with large associated air quality and climate impacts. Particulate matter from fires (or
smoke) is dominated by carbonaceous aerosol (black carbon (BC) and organic aerosol (OA)) (Akagi et al. 2011; Bond et al.
2013). As these emissions are transported through the atmosphere, they deteriorate air quality in a variety of ways. Because
of their small size and associated ability to lodge deeply in lungs, aerosols can have significant health impacts (respiratory
infections, asthma, and lung cancer) and increase cardiovascular disease (e.g., Pope and Dockery 2006 & Brook et al. 2010),
especially the high levels of PM from fire events (Reid et al. 2016). Biomass burning aerosols (BBA) can also impact the
climate system via absorbing and scattering radiation (Bond et al. 2013). In an era of increasing wildfire activity in the
western US (Westerling et al. 2006; Westerling 2016), there is a pressing need to understand how smoke from fires impacts
air quality and alters atmospheric radiation.

Globally, BB is responsible for roughly 30% of BC and nearly 90% of primary OA emissions (POA), contributing an
estimated 34 Tg yr$^{-1}$ of aerosol to the atmosphere annually (Bond et al. 2013). In addition, fires may be an important source
of secondary organic aerosol (SOA), which form from the oxidative aging of gas-phase organics emitted during combustion.
Our current understanding of SOA formation is incomplete. Recent studies demonstrate that there is no clear consensus on
the magnitude of SOA from fires, with estimates that range from virtually none to 95 Tg yr$^{-1}$ (Shrivastava et al. 2017,
Vakkari et al. 2018). Much of this spread comes from diverging results from field versus laboratory studies: the majority of
field studies have reported no secondary aerosol formation (above dilution-corrected POA concentrations; Hodshire et al.
2019) or even a decrease in OA (May et al. 2014; Liu et al. 2016; Akagi et al. 2012; Jolleys et al. 2012; May et al. 2015;
Forrister et al. 2015; Collier et al. 2016; Garofalo et al. 2019), while a few field studies observed significant SOA formation
from biomass burning emissions (Yokelson et al. 2009; Vakkari et al. 2014; Vakkari et al. 2018). Laboratory studies, to the
contrary, almost always report substantial SOA formation from fires (Grieshop et al., 2009; Hennigan et al., 2011; Ortega et



al., 2013; Tkacik et al. 2017; Lim et al. 2019). The reasons for the discrepancy across studies are not understood (Shrivastava
et al., 2017; Hodshire et al. 2019) and should be the focus of further research.

Biomass burning aerosols (BC, POA, and SOA) can have major impacts on radiation. Black carbon has a strong warming or
positive direct radiative effect (DRE) (instantaneous radiative impact), both globally and regionally, and some studies
suggest its warming direct radiative forcing (DRF) (the change in DRE from pre-industrial to present day, not including
climate feedbacks) (Heald et al. 2014) is second only to $CO_2$ (Bond et al. 2013). Black carbon from BB and gas flares also
lowers the snow and ice albedo in the Arctic, leading to additional warming (Stohl et al. 2013). Organic aerosol, because it
scatters radiation, has a negative or cooling DRE (Bond et al. 2013). It is therefore the sum of the warming from absorption
and the cooling from scattering that dictates the climate effect of BBA, leading to uncertainty in even the sign of the net
radiative effect of fires. Previous estimates of BBA DRE range from -0.01 to 0.13 W/m$^2$ (Rap et al. 2013; Ward et al. 2012).
Furthermore, when quantifying BBA impacts on radiation, differentiating anthropogenic and natural fires is important
because only the first contributes to climate forcing, or the DRF.  The uncertainty in fire radiative impacts has not been
assessed.

North America, in particular the western US, is one of the few regions in the world where more intense and frequent
wildfires have been directly tied to climate change impacts (e.g., hotter temperatures and less snowpack) (Wehner et al.
2017; Abatzogolou & Williams 2016). In addition to climate change, historical fire suppression efforts in the US have led to
increased fuel loads for fires (Marlon et al. 2012). Consequently, BBA emissions there are likely to increase in future
decades (Yue et al. 2013). Already, boreal forest fires are responsible for only 2.5% global burned area but 9% of global
BBA emissions (van der Werf et al. 2017). Biomass burning in Alaska has also accelerated in the last decade through
increases in both burned area and fire frequency leading to increases in carbon loss associated with late-season burning
(Turetsky et al. 2011). Both relative and total impacts of BB on air quality and climate forcing are expected to increase as
controls continue to reduce fossil fuel emissions and a changing climate potentially leads to more fires (Fuzzi et al. 2015;
Val Martin et al., 2015). It is, therefore, becoming increasingly important to have models and emission inventories that can
accurately characterize the impact that current and future fires and their emitted aerosols have on the environment, climate,
and human health. Several recent laboratory studies (e.g., Jolleys et al. 2014; Levin et al. 2010; McMeeking et al. 2009),
including the recent NOAA Fire Lab 2016 experiments in Missoula, MT (e.g., Koss et al. 2018; Selimovic et al. 2018; Jen et
al. 2019), have explored the BB of North American fuels, providing key constraints on smoke emissions, aging, and
properties.

Because BBA emissions cannot routinely be measured directly, a variety of global fire emission inventories have been
developed over the last decade(s) based on satellite observations. These inventories use different empirical approaches and
underlying data to represent gas and aerosol emissions from fires - each with inherent uncertainties. Aerosol emissions from



these inventories often vary by large factors depending on the region, do not agree spatially, and sometimes do not reflect
observations of concentrations and AOD well either when integrated into a model (Reddington et al. 2019; Reddington et al.
2016; Petrenko et al. 2012). In this analysis, we focus on four commonly used, but theoretically distinct inventories: the
Global Fire Emissions Database version 4 (GFED4s) (van der Werf et al. 2017) with small fires, the Fire INventory from
NCAR version 1.5 (FINN1.5) (Wiedinmyer et al. 2011), the Quick Fire Emissions Database version 2.4 (QFED2.4)
(Darmenov and da Silva, 2013), and the Global Fire Assimilation System version 1.2 (GFAS1.2) (Kaiser et al. 2012). The
two main approaches are a fire detection/burned area (FD/BA) method that relies upon burned area, which GFED4s uses, or
active fire counts, which FINN1.5 uses, and the fire radiative power (FRP) approach, which relies upon fire radiative energy
observations, an approach which both QFED2.4 and GFAS1.2 use. Comparisons among these different types of inventories
suggest that there is significant variability in the amount of dry matter burned associated with an individual active fire
detection, which is one explanation for why FD/BA and FRP inventories do not align (van der Werf et al. 2017 and
references therein). Studies using AOD to interrogate BB emission inventories give varied results but suggest that FD/BA
BBA estimates are roughly a factor of 3 too low in large BB regions (e.g., boreal North America, South America, southern
Africa, and equatorial Asia) and globally (Johnston et al., 2012; Kaiser et al., 2012; Petrenko et al., 2012; Tosca et al., 2013).
In this study we will refer to the spread across these inventories as the "uncertainty" in emissions; however, we note that
additional factors, not represented by any of these inventories, may increase the true uncertainty in the estimated emissions.
Here we use the GEOS-Chem chemical transport model and a suite of fire emission inventories to investigate the emissions
uncertainties associated with impacts of BBA on air quality and radiation. We explore the interannual and geographic
variability of fire emissions and dry matter (DM) consumed from 2004-2016 across inventories and discuss how the
uncertainty in emissions carries forward to concentrations, exposure, aerosol optical depth (AOD), and DRE with a focus on
2012 - 2014. We also explore the impact of a new model parameterization for SOA from fires.

## 2 Model and observations descriptions

### 2.1 The GEOS-Chem model

We use GEOS-Chem (www.geos-chem.org), a global chemical transport model, coupled with the rapid radiative transfer
model for global circulation models (RRTMG, Iacono et al. 2008), a configuration known as GC-RT (Heald et al. 2014), to
explore the air quality and climate impacts of BBA. GEOS-Chem is driven by assimilated meteorology from the Modern-Era
Retrospective analysis for Research and Applications, Version 2 (MERRA-2) at the NASA Global Modeling and
Assimilation Office (GMAO). We run version 12.0.0 of GEOS-Chem (https://doi.org/10.5281/zenodo.1343547) with a
horizontal resolution of 2x2.5° and 47 vertical levels with a chemical timestep of 20 minutes and a transport timestep of 10
minutes and with six month spin up simulations prior to the time periods of interest, 2012-2014 and June-July 2008. We also
perform nested simulations over North America at 0.5x0.625° (with boundary conditions from the global simulation) for



comparison against observations (IMPROVE and aircraft campaigns, see Sect. 2.3) with transport and chemistry timesteps of
5 and 10 minutes, respectively.

GEOS-Chem employs $SO_4^{2-}$–$NO_3^-$–$NH_4^+$ thermodynamics (Fountoukis & Nenes, 2007) coupled to an ozone–VOC–$NO_x$–
oxidant chemical mechanism (Mao et al., 2013; Travis et al., 2016; Miller et al., 2017) with integrated Cl-Br-I chemistry
(Sherwen et al., 2016). The model includes schemes for fine and coarse sea salt aerosols (Jaeglé et al., 2011) and mineral
dust in four size bins (Fairlie et al., 2007; Ridley et al., 2012). The standard simulation of BC in GEOS-Chem is described in
Park et al. (2003). We update this simulation per Wang et al. (2014), as follows: we update the initial hydrophilic fraction
from BB to 70% based on field observations (Wang et al., 2014 and references therein). Fossil-BC is aged from hydrophobic
to hydrophilic using the Liu et al. (2011) BC aging scheme with dynamic [OH] and [$SO_2$] per Wang et al. (2014), and
biofuel/biomass-BC is aged with an e-folding time of 4 hours. For hydrophilic BC, we use an absorption enhancement from
coating of BC of 1.1 for fossil-BC and 1.5 for biofuel/biomass-BC. We also update the BC properties for optical calculations
per Wang et al. (2014).

The standard primary organic aerosol (POA) simulation emits 50% of POA as hydrophilic and ages hydrophobic POA to
hydrophilic POA with an atmospheric lifetime of 1.15 days (Chin et al. 2002; Cooke et al. 1999). We use an organic matter
(OM) to OC ratio of 1.4 for hydrophobic OC and 2.1 for hydrophilic. The baseline model formation of SOA from BB
follows the simple scheme implemented by Kim et al. (2015) based on field results from six large campaigns summarized by
Cubison et al. (2011). This emits 0.013g SOA precursor (SOAP) per g CO emitted, which then forms non-volatile SOA on a
fixed timescale of one day. SOAP is not lost by dry or wet deposition. Recent laboratory results from the NOAA Fire Lab
2016 campaign suggest much greater SOA formation from the burning of North American fuels (Lim et al, submitted);
however, we note that, as previously discussed, uncertainties surrounding this source of SOA remain large. Based on this
study, we perform a sensitivity analysis for a new parameterization for SOA production from fires, where SOAP is estimated
as POA fire emissions scaled by a factor of 2.48. We note that this is 13 times larger than the field-based estimate of Cubison
et al. (2011), which combines the effects of POA evaporation and SOA formation (see Sect. 5 for further details).

Anthropogenic emissions (including fossil and biofuel sources) of both BC and POA follow the CEDS global inventory
(Hoesly et al. 2018) with regional inventories used when available, including NEI2011v1 over the US (Environmental
Protection Agency (EPA) National Emissions Inventory, 2015), APEI over Canada, and DICE-Africa over Africa (Marais
and Wiedinmyer 2016). Trash burning emissions are from Wiedinmyer et al. (2014). Aircraft emissions are from the AEIC
inventory (Stettler et al. 2011; Simone et al. 2013). Global annual anthropogenic emissions are 4.5 Tg yr$^{-1}$ of BC and 8.7 Tg
yr$^{-1}$ of OA in 2012. Biogenic emissions are calculated online from the MEGANv2.1 emissions framework (Guenther et al.

165  2012).




Fire emission inventories (GFED4s, FINN1.5, QFED2.4, and GFAS1.2) are specified on a daily timescale, the frequency at
which all four inventories were available. The standard version of GEOS-Chem, which we use, emits all fire emissions in the
boundary layer. Diurnal scale factors from the Western Regional Air Partnership (WRAP 2005) were applied to all
inventories per Kim et al. (2015). Additional information on each fire inventory is provided in Sect. 2.2.

We quantify simulated AOD at 550 nm, assuming that aerosols are externally mixed with a fixed lognormal size distribution
for each species and that AOD is a function of relative humidity to account for hygroscopic growth, which also varies by
species (Martin et al. 2003). Aerosol optical properties are from the Global Aerosol Data Set (GADS) database (Koepke et
al. 1997) with updates from Drury et al. (2010) and Wang et al. (2014). RRTMG calculates both longwave and shortwave
atmospheric radiative fluxes. When coupled to GEOS-Chem, this calculation is performed every 3 hours. Long and
shortwave DRE at the top of the atmosphere are summed and reported as total DRE.
**2.2 Description of fire emission inventories**
Here we describe the differences and similarities of the four fire emission inventories investigated in this study: two FD/BA
approaches (GFED4s and FINN1.5) and two FRP-based (QFED2.4 and GFAS1.2). GFED4s is the most widely used of fire
emission inventories (other inventories are sometimes scaled to it), and it employs a FD/BA approach based on the Moderate
Resolution Imaging Spectroradiometer (MODIS)-observed burned area complemented by the Carnegie–Ames–Stanford
Approach (CASA) biogeochemical model. CASA provides estimated biomass factors (i.e., combustion completeness and
fuel load) in a variety of carbon pools (e.g. leaves, grasses, litter, etc.), depending on pool-specific and environmental
conditions, which are combined with emission factors (EFs) and MODIS burned area to produce emissions (van der Werf et
al. 2017). GFED4s therefore estimates emissions as:
$M_s = A \; x \; \rho \; x \; \gamma \; x \; EF_s,$ (1)
where $M_S$ is the mass of the species of interest (g), A is burned area ($m^2$), $\gamma$ is combustion completeness (%), $\rho$ is fuel load
(kg DM/$m^2$), and $EF_S$ is the species-specific emission factor (g species/kg DM).

The fourth and most recent version of GFED (GFED4s) provides emissions at a 0.25° resolution from 1997 in near real time,
and boosts emissions to include small fires (Randerson et al. 2012). Burned area estimates from 2000 onwards are from the
MODIS MCD64A1 500m burned area maps aggregated at 0.25° resolution and a monthly time step (Giglio et al. 2013).
Because of measurement limitations, EFs, in general, are very uncertain (see Sect. 3), but GFED4s employs the most recent
compilation of EFs (Akagi et al. 2011) with some updates, such as for the temperate forest biome. GFED4s emissions are
available monthly with scalars also available to distribute emissions over daily or three-hour intervals. These scalars are only
available from 2003 onwards.



FINN1.5 follows the same FD/BA approach as GFED4s but with some differences, including: burned area is estimated from
active fire detection identified with the MODIS Thermal Anomalies Product (Giglio et al., 2006), EFs are based on the 2015
updates from Akagi et al. (2011) (http://bai.acom.ucar.edu/Data/fire/), and different land cover maps are used. FINN1.5
emissions uncertainty comes from the use of fire hot spots, assumed area burned (each fire hot spot is equivalent to 1km$^2$
burned area except grasslands, which are 0.75 km$^2$), land cover maps, biomass consumption estimates, and EFs (Wiedinmyer
et al. 2011). The original emission estimates are available at 1 km$^2$ spatial resolution and from 2002 – 2016 at both daily and
monthly mean temporal resolution. Within the GEOS-Chem model, FINN1.5 input files are available at 0.25°, and $CO_2$
emissions are produced with FINN1.5 and then other emitted species are scaled based on emission factors and land cover
type.

QFED2.4 and GFAS1.2 employ an FRP-based method, which estimates emissions using satellite observations of fire
radiative power (FRP), relying upon the following theoretical approach:
$M_s = \alpha \; x \; EF_s \; x \; FRE = \alpha \; x \; EF_s \; x \int_{t_1}^{t_2} FRP(t)dt,$                    (2)
where α is the emission coefficient (kg DM J$^{-1}$), EF$_S$ is the species-specific emission factor (g species/kg DM), and FRE in
joules (is fire radiative energy or the integral of fire radiative power (FRP in J s$^{-1}$) over time.

This FRP-based approach takes advantage of an empirically derived linear relationship between the energy released as
thermal radiation (FRE) and the mass of fuel or DM consumed during combustion (Wooster 2002; Wooster et al. 2005;
Ichoku and Kaufman 2005). This basic relationship is supported by the fact that the energy released by burning the same
amount of a fuel is similar regardless of vegetation type (Wooster et al. 2005). The energy from combustion processes not
transferred into the environment (through conductive, evaporative, and convective processes) is released as infrared
radiation, which is then assumed to be proportional to the total energy produced during combustion. One can then relate the
amount of fuel burned with the time-integrated FRE using an emission coefficient (α). In laboratory studies, the coefficient
appears to be universal, i.e. independent of fuel type (Wooster et al. 2005). For satellite-observed FRE, however, different
values are associated with different broad classes of fire types (Kaiser et al. 2012).

QFED2.4 uses the MODIS Active Fire Level 2 product (MOD14 and MYD14) and the MODIS Geolocation product
(MOD03 and MYD03) for FRP and the location of fires. A linear regression between the QFED2.4 dataset, starting with an
emission coefficient (α$_0$) from Kaiser et al. (2009), and version 2 of GFED was used to calculate the α used in QFED2.4. The
location of the fire in addition to a vegetation land type mask was used to assign the FRP to a QFED2.4 vegetation type,
which was based on an aggregated version of the International Geosphere-Biosphere Programme (IGBP) vegetation mask
with four basic classes: tropical forest, extratropical forest, savanna, and grassland. GFAS1.2 also uses the MOD14 fire
product. GFAS1.2 utilizes land cover maps based on the dominant vegetation type from GFED3 and additional organic soil





and peat maps (Kaiser et al. 2012). GFAS1.2 also derives conversion factors linking FRP and the GFEDv3.1 dry matter
combustion rates based on linear regressions between the two.

QFED2.4 and GFAS1.2 utilize EFs from Andreae and Merlet (2001). QFED2.4 scales its aerosol emissions to better
represent MODIS-observed AOD, using biome-dependent strength factors. It should be noted that these enhancement factors
were based on the GEOS model, and depend on the underlying model configuration, most importantly, the single assumed
OM:OC ratio of 1.4, but also the specific anthropogenic emissions and the radiative properties of aerosols in the model.
Thus, these enhancement factors that scale to AOD could differ substantially in a model that treats these factors differently.
To our knowledge, these differences have not been accounted for in previous model studies that have used QFED (e.g., Kim
et al. 2015; Marais et al. 2016; Lu et al. 2015; Saide et al. 2015; Zhang et al. 2014). We make no effort to re-derive the
biome-specific enhancement factors for GEOS-Chem. In an effort to ensure that global totals of emitted BC and OA are
consistent with those reported by QFED2.4, we scale down emissions by a uniform factor of 0.69 (1.4/average OM:OC ratio
in GEOS-Chem in 2012). QFEDv2.4 provides daily mean emissions and is available at 0.1° resolution from 2003 – 2016.
GFAS1.2 provides daily mean emissions and is available from 2003 – 2019 at 0.1° resolution.

Some advantages of QFED2.4, GFAS1.2, and other FRP-based inventories are that the uncertain factors used in FD/BA
inventories to convert burned area to DM consumed (fuel load and combustion completeness) can be bypassed, and that FRP
observations are more sensitive to small fires than burned area observations (MODIS has detection limits of ~5MW and
$50m^2$, respectively). On the other hand, active fire observations (both active fire counts and FRP) can only detect fires during
the burning phase, while the accumulated burned area can be detected for an extended period of time after the burning phase.
FRP-based emission estimates therefore contain errors due to assumptions on undetectable fire activity under cloud cover
and between satellite overpasses (for low-earth orbiting instruments like MODIS). Smouldering and peat fires are difficult to
quantify with both methods: FRP-based approaches suffer from weak thermal signatures and uncertain emission coefficients
(Darmenov and da Silva 2013), and FD/BA-based approaches suffer from missing information on burn depth and thus
combustion completeness.
**2.3 In-situ observations**
The ARCTAS (Arctic Research of the Composition of the Troposphere from Aircraft and Satellites) summer airborne
campaign surveyed large swaths of the Arctic with an emphasis on probing forest fire smoke plumes using the NASA DC-8
aircraft from June 18 to July 13, 2008 (Jacob et al. 2010) (see Fig. 1 for flight tracks). Black carbon mass concentrations
were measured with a single particle soot photometer (SP-2, Schwarz et al. 2008). For ARCTAS, the SP-2 detection range
for particle diameter is 80-860nm, and the uncertainty is estimated to be 10% (Kondo et al. 2011). Organic aerosol was
measured using a high-resolution time-of-flight aerosol mass spectrometer (CU-Boulder Aerodyne HR-ToF-AMS, DeCarlo
et al. 2006; Canagaratna et al. 2007; Cubison et al. 2011) with a 2σ estimated uncertainty of 38% for OA (Bahreini et al.





2009) and a size detection limit extending down to 35nm vacuum aerodynamic (about 25 nm geometric diameter for typical
BBOA densities) (DeCarlo et al. 2006; 2008). Concentration detection limits for OA for 1 min. data are ~0.16 μg m$^{-3}$
(DeCarlo et al., 2006; Dunlea et al., 2009), several orders-of-magnitude lower than typical field BBOA concentrations.
Acetonitrile, a useful tracer for BB, was measured using a Proton-Transfer-Reaction Mass-Spectrometer (PTR-MS, Hansel
et al. 1995; Wisthaler et al. 2002) and used as a filter to help isolate BB influence.

Observations from the Deep Convective Clouds and Chemistry (DC3) campaign are also included in our analyses. DC3
focused on thunderstorms and their impact on the chemical composition of the troposphere and also documented BB plumes
and their interactions with deep convection in the Southern Great Plains, the Colorado Front Range, and the southeastern US.
Flights occurred from May 18 to June 22, 2012 (Barth et al. 2015) (Fig. 1). As in ARCTAS, BC was measured using the SP-
2, and OA was measured using an HR-ToF-AMS. The detection range for BC mass from the SP-2 corresponds to 90-550 nm
volume equivalent diameter, assuming 1.8 g cm$^{-3}$ density, with ± 30% total uncertainty in the accumulation mode BC mass
mixing ratio (Schwarz et al. 2013). Acetonitrile was again measured using a PTR-MS (Hansel et al. 1995; Wisthaler et al.
2002). For comparison with airborne measurements, the model was sampled to the nearest grid box both temporally and
spatially to each flight track using 1-minute aircraft data. We then average both the model and the observations to the model
grid box.

As the spatial and temporal coverage of aircraft campaigns is limited, we also include surface observations from 168 sites in
the contiguous United States (CONUS) that are part of the IMPROVE aerosol network (Interagency Monitoring of Protected
Visual Environments, http://vista.cira.colostate.edu/improve/) from 2012 and compare against 24-hour averaged model
results. Black carbon and OC are measured using a PM$_{2.5}$ size-selective filter-based thermal method in this network (Chow et
al. 2007). We use a conversion factor of 1.8 from OC to OA mass (Malm and Hand 2007), which is the average of fresh and
more aged OA in the model, to represent average surface conditions (note that the same OM:OC is applied to the model
simulation when compared against IMPROVE).

### 2.4 MODIS AOD observations

Aerosol optical depth (AOD), the column total aerosol extinction, is directly proportional to the total mass concentration of
aerosol in an atmospheric column (Levy et al., 2007, 2010) and is commonly measured by satellites. AOD measurements
capture all aerosol contributions and, therefore, do not provide a unique quantitative constraint on BBA, but they can be a
used to understand spatial and interannual BB patterns.

We use the MODIS Collection 6 level 3 daily product of satellite AOD retrievals at 550nm and 10km resolution (Levy et al.
2013 & Sayer et al. 2014) from the Aqua platform and re-grid MODIS AOD from 1x1° to the model grid of 2x2.5° for
further comparison with GEOS-Chem AOD. AOD retrievals from Aqua are used because the cross-over time of Aqua (early



afternoon) typically coincides with peak burning activity and a well-mixed boundary layer. We use a merged AOD product
(Dark Target-Deep Blue Combined Mean) from the Collection 6 MODIS data that combines ocean and vegetated land
surface retrievals (Dark Target) and bright land surface retrievals (Deep Blue) to maximize coverage. Retrieved AOD ($\tau$) is
estimated to be accurate to $\pm0.03 \pm 0.05\tau$ over the ocean (Remer et al. 2005), to $\pm0.05 \pm 0.15\tau$ over dark land surfaces (Levy
et al. 2010), and to $\pm0.05 \pm 0.20\tau$ over bright surfaces (Hsu et al. 2006; Sayer et al. 2013). The model was sampled at the
satellite overpass time (1330 local time). In addition, we filter out AOD values from both MODIS and the model for which
the cloud fraction from MODIS is greater than 80% to eliminate potential cloud contamination.

## 3 Underlying emissions and dry matter uncertainty

Figure 2 demonstrates the large differences in total annual BBA emissions estimated by the four different fire emission
inventories from 2004-2016 for boreal North America (BONA, Canada and Alaska), the contiguous US (CONUS), and the
globe. Emission totals over other large BB regions that are not the focus of this study (Amazon, Africa, and Asia) are shown
in Fig. S1. We focus on BC and OC (note that inventories provide OC, not OA) emissions in our analysis, but also provide a
summary of CO for context, which generally follows the trends observed for OC (as does $NO_x$, not shown). Globally,
emissions of BC and OC are highest in QFED2.4 (3.1Tg yr$^{-1}$ and 28.3Tg yr$^{-1}$, respectively) but emissions are also most
variable in this inventory (i.e., more variability from 2004-2016 as evidenced by the taller boxplots) (Fig. 2). Average global
annual emissions are smallest in GFED4s for BC, and, for OC and CO, FINN1.5 emissions are smallest – though very
similar to GFED4s for OC and similar to QFED for CO. Global mean BC emissions differ by roughly a factor of 2.3 across
the inventories while mean annual OC emissions differ by less (~ a factor of 1.7). The inventories show a smaller range in
mean CO emissions (~ a factor of 1.1): from GFAS1.2 (360Tg yr$^{-1}$) to FINN1.5 (327Tg yr$^{-1}$).

The spread in BBA emissions across North America is larger than that seen globally. In BONA, mean annual BC and OC
emissions show a factor of roughly five and four range, respectively, from the smallest, FINN1.5 (0.02Tg yr$^{-1}$ and 0.4Tg yr$^{-1}$,
respectively), to the largest, GFAS1.2 (0.1Tg yr$^{-1}$ and 1.7Tg yr$^{-1}$, respectively). The relative magnitudes of the four
inventories are consistent across species for CONUS with QFED2.4 largest (0.09Tg yr$^{-1}$ and 1.3Tg yr$^{-1}$, for BC and OC
respectively), followed by GFAS1.2 (0.04Tg yr$^{-1}$ and 0.5Tg yr$^{-1}$, for BC and OC respectively), and then FINN1.5 (0.03Tg yr$^{-1}$
and 0.2Tg yr$^{-1}$, for BC and OC respectively) and GFED4s (0.01Tg yr$^{-1}$ and 0.3Tg yr$^{-1}$, for BC and OC respectively) – where
the exception is that the mean OC emissions from GFED4s are slightly larger than those of FINN1.5. The range of values is
very similar for BC and OC in CONUS (a factor of ~7 for BC and ~6 for OC).

Multiple studies (e.g., Akagi et al. 2011; Alvarado et al. 2010; Urbanski et al. 2011) have identified uncertainties in EFs as a
large source of uncertainty in BB emissions. Table 1 confirms that there are large differences in the EFs used in the four
inventories explored here in North America, particularly in boreal and agricultural regions. For example, OC boreal forest





EFs range from 7.8 to 9.6 g/kg DM and BC from 0.2 to 0.56g/kg DM. The EFs used in each inventory are shown spatially
over North America in Fig. 3. The uncertainty in EFs is associated with: measurement technique, variation in the
experimental conditions used to measure species' EFs in a laboratory, post-processing and aging that can change smoke
composition rapidly but is likely not yet fully mechanistically understood, and poorly characterized combustion and fire
types (Akagi et al. 2011). Measured EFs vary considerable from different fuels (Jolleys et al. 2014; McMeeking et al. 2009);
however, only coarse vegetation types (e.g., boreal forests) are typically delineated in emission inventories, making it
difficult to apply laboratory-measured EFs. Of relevance to this study, relatively few measurements of BB have been made
in temperate regions, such as large portions of the US, where much of the BB is prescribed for land management but
controlled to protect air quality (Akagi et al. 2011), conditions which may lead to substantially lower BBA emissions (Liu et
al., 2017). Another potential source of uncertainty in EFs is that experimentally-derived OC EFs may represent SOA as well
as POA; EFs presented in compilations (Akagi et al. 2011; Andreae and Merlet 2001) are generally calculated from fresh
smoke where the quantity of SOA production is not well constrained.

We quantify how the range in EFs contributes to the overall spread in BBA emissions.  First, we divide emissions by the
applied EFs to estimate the underlying dry matter (DM) consumed across inventories in the same regions and years as our
emissions analysis (Fig. 4) to isolate the importance of EFs. We note that the two-FD/BA inventories (GFED4s and
FINN1.5) quantify DM consumption in the construction of the inventory; however, for the FRP-based inventories (QFED2.4
and GFAS1.2) this division results in an effective DM consumed (FRE multiplied by an emission coefficient). We show DM
calculated from BC emissions except for QFED2.4, where we use the effective DM calculated from the CO emissions so as
to avoid any confounding issues with the aerosol strength scaling factors discussed briefly in Sect. 2.2. Across all regions,
the range in DM tracks very closely the range observed across emissions, suggesting that the uncertainty in the underlying
DM, not EFs, is the predominant factor in emissions uncertainty. We note that the large range in the DM consumed globally
alongside the similar global CO emissions indicates that large differences in the EFs of CO and different vegetation
classifications offset the DM differences for this species.

To further illustrate the role of EFs, Fig. 5 shows the time series of total annual emissions from 2004-2016 for GFED4s,
alongside the estimated emissions obtained by applying the GFED4s EFs to the estimated DM for the other three original
inventories (applied using each inventories' respective vegetation mask). We then compare total annual emissions from the
original inventories (dashed lines) with their GFED4s-EF counterparts (solid lines) and with the original GFED4s inventory
from 2004-2016 (Fig. 5). While eliminating the variation in assumed EFs does constrict the range in emissions across the
inventories across North America and globally, there remain substantial differences. This suggests that EFs are important but
that underlying DM burned is the largest source of fire emissions uncertainty – consistent with previous work (Van Leeuwen
et al. 2014). One reason for this is that substantial uncertainties are associated with using biome-averaged values to represent



DM consumed for whole biomes (Veraverbeke et al. 2015; Van Leeuwen et al. 2014) and that satellite products and
assumptions used to capture fuel burned vary significantly (van der Werf et al. 2017 and references therein).

Furthermore, assuming that the EFs used in the four inventories are all equally reasonable values, we can estimate a much
larger range in plausible fire emissions by multiplying the minimum and the maximum DM consumed across the inventories
by the smallest and largest EFs (Table 1) using the GFED4s vegetation mask. Globally, this calculation suggests a plausible
range that spans a factor of 24 for BC and 18 for OC compared to the inventory spread of 2.3 and 1.7, respectively. This
suggests that using the range across these four inventories may be a modest estimate of the uncertainty in fire emissions.

Interannual differences, especially in North America, are fairly consistent across the inventories except for 2014 (Fig. 5)
where QFED2.4 trends down while the other three increase. It should be noted that an updated version of QFED (v2.5r1)
does not show this decreasing trend in 2014. Globally and in CONUS, GFED4s, GFAS1.2, and QFED2.4 show similar
interannual differences while FINN1.5 shows the greatest interannual variability and different maximum and minimum
years. We note that 2012 is a fairly typical fire year (see Fig. 5), and much of the following analysis will focus on this year.

We also explore the seasonality of BC and OC emissions represented in the inventories for BONA, CONUS, and globally
across the same 13 years (Fig. 6). The seasonality, including relative magnitude, is generally consistent across regions and
species. Some seasonal features (e.g., the October-November enhancement in BONA and the springtime enhancement in
CONUS) are only visible in the three inventories that rely on active fire counts or FRP – FINN1.5, QFED2.4, and GFAS1.2
– which is consistent with work suggesting that these methodologies pick up small fires better than GFED4s (Kaiser et al.
2012). The fall peak in the boreal region is driven by fires in eastern British Columbia. The seasonal CONUS springtime
peak is primarily associated with small fires (as identified in GFED4s), likely linked to agricultural burns.
**4 How emissions uncertainty impacts mass concentrations and AOD**
Given the large range in fire emissions, we use observations to try to assess which, if any, inventory is most realistic. We use
IMPROVE surface observations and two airborne campaigns to compare with model simulations driven by each inventory.
As another constraint on aerosol abundance, we also compare model AOD with MODIS-observed AOD in North America.

We test the model against IMPROVE observations of surface concentrations across the US and find significant variation in
model skill across the inventories with QFED2.4 generally biased high and FINN1.5 low (Fig. 7 & 8). Seasonal comparisons
of IMPROVE surface concentrations with simulated concentrations driven by the four different inventories show similar
patterns across aerosol species but significant differences between the western and eastern US (Fig. 7). This is likely related
to how well the inventories capture the differences in burning regimes in the western (predominantly wildfires) and eastern



(mostly prescribed and agricultural burns) US (Brey et al. 2018). The southeastern US, in particular, is of interest to the
public health and policy communities because a prevalence of agricultural burning there, which dominates burned surface
area (Nowell et al. 2018), may have a stronger impact on low altitude air quality in a relative sense than large wildfires that
inject higher into the air. We also analyse the western and eastern US separately because, in the east, the magnitude of fire
emissions is lower and BC, in particular, is dominated by anthropogenic sources. In the western US, GFED4s and GFAS1.2-
driven concentrations of both BC and OA match the seasonality and magnitude of IMPROVE observations well. QFED2.4 is
biased high, particularly during the peak in the wildfire season (August-September). FINN1.5-based concentrations are
biased low and are virtually indistinguishable from simulations with no BB. In the eastern US, because fire is a smaller
relative source of carbonaceous aerosol, there is less of a spread between the simulations. All inventories other than
QFED2.4 do a reasonable job capturing observations with a general tendency for simulated BC and OA to be a bit too high,
suggesting an overestimate in anthropogenic emissions in the eastern US. However, the $25^{th}$ to $75^{th}$ percentile bars on the
observations show that across the US for BC and in the west for OA, virtually all the simulations fall within this range of the
measurements. QFED2.4 overestimates OA well beyond the $25^{th}$ to $75^{th}$ percentile range in the eastern US, starting with the
northern hemispheric wildfire season in May and continuing the overestimate through the end of the calendar year.
Figure 8 illustrates the ability of these simulations to capture the spatial distribution of observed surface concentrations
during the fire season (May-September). Similar skill is seen across both aerosol species for GFED4s and GFAS1.2, but
FINN1.5 matches observed BC somewhat better than OA and QFED2.4 matches OA somewhat better than BC. Consistent
with the seasonal IMPROVE analysis, simulations driven by GFED4s, QFED2.4, and GFAS1.2 have greater skill in the
western US than the eastern US while the FINN1.5-driven simulation performs better in the east. QFED2.4 is generally
biased high, especially in the Pacific Northwest and, to some extent, in the southeastern US. However, QFED2.4 also has the
highest skill in reproducing the spatial patterns of the highest concentrations when compared against the $95^{th}$ percentile of
observed concentrations (not shown).
The ability of models to accurately represent aerosol concentrations aloft is also important for both air quality and climate,
and we use two fire-influenced aircraft campaigns, DC3 and ARCTAS, to explore the model skill in this dimension. These
campaigns provide observations from two very different fire regimes across North America (See Sect. 2.3) – DC3 in the
central/southeastern US and a subset of ARCTAS focusing on boreal Canada. In addition to median vertical profiles for both
BC and OA for each campaign, we also show median vertical profiles filtered by the top $25^{th}$ percentile of acetonitrile
(equivalent to a concentration cut off of 167 ppt for DC3 and 213 ppt for boreal ARCTAS), a useful BB tracer that allows us
to investigate the most BB-influenced data.
We find that concentrations driven by the various inventories perform somewhat differently against each of the campaigns
(Fig. 9 & 10). Across both campaigns, QFED2.4-driven modelled concentrations are generally biased high, particularly





towards the surface, while FINN1.5 simulations are nearly always biased low (Fig. 9 & 10). QFED2.4 has been constrained
to observed AOD, so one could assume that it would perform best. We find that after adjusting the QFED2.4 emissions
downward to account for our different OM:OC ratio, QFED2.4 simulations of OA do match observed concentrations fairly
well; however, BC concentrations remain much too high. This suggests that the QFED2.4 biome-specific adjustment factors
should not be applied to BC and that the scaling factor applied in this inventory to match AOD constraints may be
accounting for errors in other properties (i.e. optical properties or background aerosol), not fire emissions. This is consistent
with recent work showing that even when observed and modelled concentrations agree in the Amazon, observed and
modelled AOD sometimes do not (Reddington et al., 2019). Over the continental US (Fig. 9) QFED2.4 emissions result in
the highest concentrations of OA and BC; however, in the boreal region (Fig. 10), simulations driven by GFAS1.2 (as well
as GFED4s to a lesser extent) produce more smoke than QFED2.4, consistent with the relative emissions magnitudes show
for these regions in Figures 2 & 5. As a result, both GFAS1.2 and GFED4s significantly overestimate both BC and OA
concentrations towards the surface in the boreal region.

In DC3, all four inventories, and even the noBB run, overestimate the BC median vertical profile, suggesting that
anthropogenic emissions are overestimated in the southeastern US, consistent with the IMPROVE analysis. This is
reinforced by the DC3 BC vertical profile filtered for fire influence where three of the inventories (GFED4s, FINN1.5, and
GFAS1.2 to a lesser extent) match observations quite well. Similarly, in boreal ARCTAS, all the inventories but FINN1.5
overestimate BC concentrations, especially towards the surface.

This analysis suggests that anthropogenic emissions of BC may be overestimated throughout the U.S., that the two FRP-
based inventories and GFED4s, to some extent, may overestimate boreal emissions, and that FINN1.5 emissions are too low
throughout, but particularly in boreal regions. In concert with the analysis at IMPROVE sites, this indicates that GFED4s-
driven simulations generally provide the best match to observations, but with substantial under/over-estimates in some
regions and species.

Our comparisons with in situ mass concentrations, both at the surface and aloft, consistently suggest that the FINN1.5
inventory substantially underestimates fires over North America. Scaling relationships between fire activity and dry matter
consumed should be re-visited for this inventory for North American fuels. One likely cause of the underestimation of North
American fires by FINN1.5 is that the MODIS Land Cover Type (LCT) data used to define burned ecosystems assigns
shrubs where other classifications assign forest, leading to lower fuel burned estimates. A second likely contributor to this
underestimate is that the way in which burned area is calculated from active fire counts underestimates large wildfires, which
is particularly relevant for the western US. This underestimation was also seen in earlier work by Pfister et al. 2011, using
FINN1.5 to explore CO from fires in California.



Some of the disagreement aloft with the baseline model across inventories may be related to the model failure to capture
injection heights for some fires which loft aerosols above the boundary layer. This is not represented in the simulations
shown here, but typical approaches put too much aerosol at the top of boundary layer (~2km) (Zhu et al. 2018) (See Fig. S3
for an injection height sensitivity test). It is also worth noting that sampling in the DC3 campaign was biased towards
convective outflow given campaign goals, and it is possible that the model may also have errors in convection and
convective removal.
Figure 11 shows the spatial distribution of average AOD over North America during the northern hemispheric fire season
(May – September) in both 2012 and 2014 compared to MODIS-observed AOD. In general, the model simulation
underestimates observed AOD, which may result from a combination of errors in model optics, background aerosol, or cloud
contamination in the MODIS product. We note that Reddington et al. (2019) similarly show that their model underestimates
AOD, even when it captures the observed mass concentrations of PM over the Amazon. Here we focus on the fire-driven
AOD features.  Across both years, FINN1.5 AOD is low compared to MODIS in CONUS and does not capture the fires in
BONA. GFED4s and GFAS1.2-driven AOD look quite similar to each other across years and include the large fire
signatures in BONA that MODIS observes. AOD driven by QFED2.4 identifies the boreal, and potentially Pacific
Northwest, fire signatures in 2012 but misses the large boreal hot spot in 2014 that is evident in both MODIS-observed and
GFED4s and GFAS1.2 AOD.
**5 Secondary organic aerosol from biomass burning and its implications**
Previous simulations in Sect. 4 included the GEOS-Chem default minor source of SOA from fires. The recent NOAA Fire
Lab 2016 experiment (Lim et al., 2019) reported large increases in OA mass when fire emissions were oxidatively aged, as
have many other laboratory studies; though, this has not been observed in the majority of field campaigns (see Sect. 1).
While uncertainties on this potential source of additional OA mass are large, we test the sensitivity of our results to this
additional source.  The default scheme ((0.013 times CO emissions) (Kim et al. 2015; Cubison et al. 2011)) results in a mean
annual global source of BB SOA (~5 Tg yr$^{-1}$) from GFED4s, which is at the lower range of potential annual global fire SOA
source amounts reported in Shrivastava et al. (2017). We implement a new parameterization from the NOAA Fire Lab 2016
lab studies for SOA production from BB based on Lim et al. (submitted) (2.48 times POA emissions). This new scheme
produces a mean annual global GFED4s source of BB SOA of ~41 Tg yr$^{-1}$, which is roughly in the middle of estimates
reported in Shrivastava et al. 2017. In principle, such a large additional source of OA should be distinguishable from
observations. However, our previous analysis using the default scheme demonstrates that the range in estimated POA is so
large that it is challenging to say how much additional OA mass from SOA from BB would be consistent with the
observations. In particular, even with negligible SOA the model already matches observed OA with at least one inventory
(QFED2.4). With this new parameterization, we show a roughly order of magnitude increase in the BB SOA burden (and





thus more than a doubling of total OA) from GFED4s in 2012 with similar increases across the other inventories. Figure 12
shows how this new SOA impacts model-observation agreement with the DC3 and ARCTAS campaigns. The QFED2.4
simulations now overestimate OA across campaigns while FINN1.5 simulations improve against observations modestly,
consistent with its smaller BB OA burdens to start with. It is possible that the AOD-based scaling of QFED2.4 emissions
previously compensated for underestimated SOA. With the new SOA parametrization, GFED4s and GFAS1.2 simulations
are better able to capture the magnitude of the mean concentrations observed during DC3.  However, for boreal ARCTAS,
GFED4s and GFAS1.2-driven simulations with the default scheme captured observed OA concentrations and indeed
overestimated (Fig. 10); thus, this new large source of fire SOA exacerbates this overestimate. Our analysis of observations
over North America can neither preclude nor confirm the presence of a large source of SOA from fires, given the uncertainty
in POA emissions over the region. This additional SOA source is not included in the assessment of air quality and radiative
impacts of fires in Sections 6 and 7.
**6 How emissions uncertainty translates to air quality and fire PM exposure**
We next explore how uncertainty in fire emissions affects estimates of air quality impacts. We show the differences in fire
$PM_{2.5}$ (PM under 2.5 microns) exposure spatially (Fig. S4) and quantify the range in population-weighted fire $PM_{2.5}$ exposure
in 2012 across North America (Canada and CONUS only) given by the four inventories. We calculate fire $PM_{2.5}$ exposure by
averaging surface concentrations of the sum of BC and OA from BB across North America in 2012. We then calculate
population-weighted annual fire $PM_{2.5}$ for each inventory by using population data from the Gridded Population of the
World, Version 4 (GPWv4), created by the Center for International Earth Science Information Network (CIESIN) and
available from the Socioeconomic Data and Applications Center (SEDAC) (Accessed 6 February 2019). We linearly
interpolate the gridded UN-adjusted population count dataset, which has a native resolution of 30 arc-seconds and provides
population estimates for 2000, 2005, 2010, 2015, and 2020, to 2012 and grid the data to the GEOS-Chem nested grid
(0.5x0.625°). Figure 13 shows that the range in BBA emissions carries forward to uncertainty in 2012 North America fire
$PM_{2.5}$ exposure with a range of $0.5 - 1.6 \mu g\ m^{-3}$. The World Health Organization (WHO) air quality guidelines for annual
mean $PM_{2.5}$ are $10\ \mu g\ m^{-3}$, and the US EPA annual standard for $PM_{2.5}$ is $12\ \mu g\ m^{-3}$. Thus, the range in fire $PM_{2.5}$ exposure
across the inventories in North America is equivalent to roughly 10% of these air quality standards. In addition, the
population-weighted mean $PM_{2.5}$ exposure due to fires in North America varies by about a factor of two between different
years, reflecting the location and intensity of different fire events (see Fig. S5 and S6 for an analysis of $2012 - 2014$ at
2x2.5°).



## 7 Impacts on the direct radiative effect

Across North America and globally, we compare the mean annual top-of-atmosphere (TOA) all-sky DRE of BB-only BC and OA driven by each of the inventories with the OA DRF reported in the Fifth Assessment Report (AR5) of the Intergovernmental Panel on Climate Change (IPCC). We quantify the annual mean BBA DRE in 2012 (Fig. 14) and the Northern Hemispheric fire season (May – September) average DRE in each year from 2012 to 2014 (Fig. S7) to investigate interannual variability. The differences across inventories seen in the sections above translate to the large ranges in DRE estimated for BONA and CONUS with smaller, but still significant, ranges seen globally.

For 2012, GFAS1.2-driven global DRE is largest in absolute magnitude for BBA (-0.11 W/m$^2$) with FINN1.5 smallest (-0.048 W/m$^2$) (See Table S1 for underlying values). These values are significantly more negative than previous estimates of BBA DRE, which ranged from -0.01 to 0.13 W/m$^2$ (Rap et al. 2013; Ward et al. 2012). Previous work suggests that the whitening of fire-generated brown carbon (BrC) may limit the global absorption from BrC (Forrister et al., 2015; Wang et al., 2016). Wang et al. (2018) estimate a modest global mean DRE of BrC of +0.048 Wm$^{-2}$ when accounting for this whitening; however, uncertainties on the magnitude and the evolution of absorption of BrC remain large. We treat OA as scattering here, which may lead to a positive bias in the total DRE of carbonaceous aerosol from smoke, thus we focus on the range in values associated with the use of various fire inventories rather than the absolute magnitude of the DRE. The range across the 2012 annual global mean inventory-driven BBA DRE is -0.062 W/m$^2$, which is comparable to the magnitude of the direct radiative forcing of OA (-0.09 W m$^{-2}$) reported in the in AR5 (IPCC 2013). Only some fires contribute to the DRF, but we have shown here that the uncertainty in BBA DRE as represented by the spread in values driven by different inventories is on a comparable scale to the anthropogenic influence on OA forcing. While we have not assessed the annual global mean BBA DRE across other years, we have quantified the northern hemispheric fire season BBA DRE from 2012-2014, which show generally similar trends across years with some variability; larger boreal fire years generally affect the DRE driven by GFED4s and GFAS1.2 the most (see 2014 in Fig. S7). 2014 also appears to be an outlier year where GFED4s and GFAS1.2-driven OA DRE is larger than QFED2.4-driven DRE across both BONA and CONUS and also globally, consistent with our emissions analysis (See Fig. 5).The IPCC estimate of aerosols' contributions to the DRF only includes one set of historical fire emissions and one for each RCP – this choice allows for better intermodal comparisons but masks underlying uncertainty from fire emissions, which we have shown here to be important.

## 8 Conclusions

Most models do not test basic uncertainty associated with fire emissions both in air quality and climate studies – our work suggests that this uncertainty is large and may substantially impact our understanding of fire impacts. We provide an evaluation of this uncertainty by comparing multiple, commonly-used fire emission inventories (GFED4s, FINN1.5, QFED2.4, and GFAS1.2) that have become available in the last five to ten years. We show that the four inventories perform





differently depending on species, location, and season. We also calculate that average BC and OC emissions differ by
roughly a factor of five and four, respectively, across the inventories in BONA. The range in BC and OC emissions in
CONUS is even larger (a factor of ~7 and 6, respectively). Global ranges in BC emissions are smaller than those in North
America (~2.3) with a somewhat more modest spread (~1.7) in OC emissions, possibly because of emission factor
differences. We also show that dry matter, not emission factor, differences are the driving force for emissions variation
across inventories.

With such large differences in emissions, we test which of these inventories drives model simulations closest to observations
over North America. We show that modeled concentrations both at the surface and aloft show variable skill across
inventories when compared to in situ observations (IMPROVE, DC3 and ARCTAS campaigns) with FINN1.5 biased low for
BC and OA and QFED2.4 biased high against observed BC. GFED4s and GFAS1.2-driven AOD also do a better job
matching MODIS-observed AOD over the regions, in general and with specific features, than FINN1.5 and QFED2.4.
QFED2.4 emissions may be biased high because they were scaled up to ensure that the GEOS model AOD simulation
matches satellite-observed AOD, potentially mis-attributing biases in aerosol extinction efficiency and SOA formation in the
GEOS model to emission; MODIS AOD has also been shown to be high in some environments (Lapina et al. 2011). That
these enhancement factors are too high is further reinforced by the fact that, after adjusting the QFED2.4 emissions
downward to account for our different OM:OC ratio, QFED2.4 simulations of OA match observed concentrations fairly well
across campaigns – while BC concentrations remain much too high. The assumptions that FINN1.5 uses to compute burned
area from active fire counts likely contribute to its low bias and should be revisited, especially for regions with large
wildfires (e.g., boreal Canada and the western US). We also show that a laboratory-based parameterization for fire SOA,
scaled from fire POA emissions, does improve model agreement with observations in some regions. However, from our
comparisons, the range in POA emissions makes it challenging to discern whether SOA from fires is significant.

This range in fire emissions also carries through to uncertainties in the air quality and radiation impacts of fires, which we
have shown to be large and significant. Over North America depending on the inventory used, large differences in both the
spatial extent of BBA-only annual surface concentrations and also in population-weighted annual fire PM2.5 exposure (0.5 -
1.6 µg m$^{-3}$ for 2012) arise. We have also shown that fire emissions uncertainty produces a considerable envelope in global
BBA DRE (-0.062 W m$^{-2}$), roughly comparable to the direct radiative forcing of OA (-0.09 W m$^{-2}$) reported in AR5.

Additional evaluation of satellite-based fire emission inventories, particularly in other large BB source regions, would help
to provide insight into fire emissions uncertainty. Observations at all scales (surface, aloft, and satellite) are needed to better
constrain our understanding of fire emissions and processing. To bridge fire emissions and subsequent impacts, additional
investigation of uncertainties in fire aerosol aging and processing (e.g., injection heights, mixing state, SOA formation, etc.)



is needed. Our work suggests that emissions uncertainty is a major factor in our ability to model the air quality and climate
impacts of fires and should be incorporated into modeling studies of both.
**9 Supplement link**
[To be added by Copernicus]
**10 Author contribution**
CLH and TSC formulated the research question and wrote the paper. TSC performed modelling and analysis. CW, AD, AD,
and JK developed the FINN1.5, QFED2.4, and GFAS1.2 emission inventories used here and provided input on the
manuscript. JLJ, PCJ, YK, NM, and JS made measurements of carbonaceous aerosol mass concentrations during DC3 and
ARCTAS used in this analysis and provided input on the manuscript.
**11 Competing interests**
The authors declare no conflicts of interest.
**12 Acknowledgements**
This study was supported by the NOAA Climate Program Office (grant NA16OAR4310112) and an Ida M. Green
Fellowship (MIT) to TSC. PCJ and JLJ were supported by NASA 80NSSC18K0630.

The authors thank the primary developers of GFED4s: Guido van der Werf, James Randerson, and Louis Giglio. The
authors also thank Mat Evans and Killian Murphy for early versions of processed GFAS1.2 files; Xuan Wang and Katherine
Travis for useful discussions regarding the GEOS-Chem simulation; Jesse Kroll, Chris Cappa, and Chris Lim for early
discussion of their SOA parameterization from the NOAA Fire Lab 2016 study; and Armin Wisthaler and Tomas Mikoviny
for acetonitrile measurements from both ARCTAS and DC3.





We acknowledge data from the IMPROVE network. IMPROVE is a collaborative association of state, tribal, and federal
agencies, and international partners. US Environmental Protection Agency is the primary funding source, with contracting
and research support from the National Park Service. The Air Quality Group at the University of California, Davis is the
central analytical laboratory, with ion analysis provided by Research Triangle Institute, and carbon analysis provided by
Desert Research Institute.

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






| Emission factors across inventories and vegetation types (g species/kg dry matter) | | | | | | | | |
|---|---|---|---|---|---|---|---|---|
| | BC | | | | OC | | | |
| Types: | GFED4s | FINN1.5 | QFED2.4[AM] | GFAS1.2[AM] | GFED4s | FINN1.5 | QFED2.4[AM] | GFAS1.2[AM] |
| temp forest | 0.5[AM] | 0.56[An] | 2.52 | 0.56 | 0.56 | 9.6[AM] | 7.6[An] | 28.38* | 9.14 | 9.1 |
| boreal forest | 0.5[AM] | 0.2[Mc] | 2.52 | 0.56 | 0.56 | 9.6[AM] | 7.8[Mc] | 28.38* | 9.14 | 9.1 |
| sav, grass, shrub | 0.37[Ak] | 0.37 (SG)/ 0.5 (WS)[Ak] | 0.86 | 0.48 | 0.46 | 2.62[Ak] | 2.62 (SG)[Ak]/ 6.6(WS)[Mc] | 4.22* | 3.40 | 3.2 |
| tropical forests | 0.52[Ak] | 0.52[Ak] | 1.65 | 0.66 | 0.57 | 4.71[Ak] | 4.71[Ak] | 8.97* | 5.20 | 4.3 |
| ag | 0.75[Ak] | 0.69[AM] | -- | -- | 0.42 | 2.3[Ak] | 3.3[AM] | -- | -- | 4.2 |

**Table 1: Emissions factors used in each inventory. Superscripted AM is from Andreae and Merlet 2001, Ak is from Akagi et al.**
**2011, An is Andreae and Rosenfeld 2008, and Mc is McMeeking et al. 2009. Note that QFED2.4 and GFAS1.2 EFs shown here for**
**BC and OC are entirely from Andreae and Merlet 2001. *The first QFED2.4 column shows the underlying EFs (shown in the**
**second QFED2.4 column) multiplied by their biome-specific enhancement factor. We also adjust this factor down by the ratio of**
**1.4 (the OM:OC ratio used in the GEOS model) to the average OM:OC ratio use in GEOS-Chem in 2012 (see Section 2.2 for**
**details).**






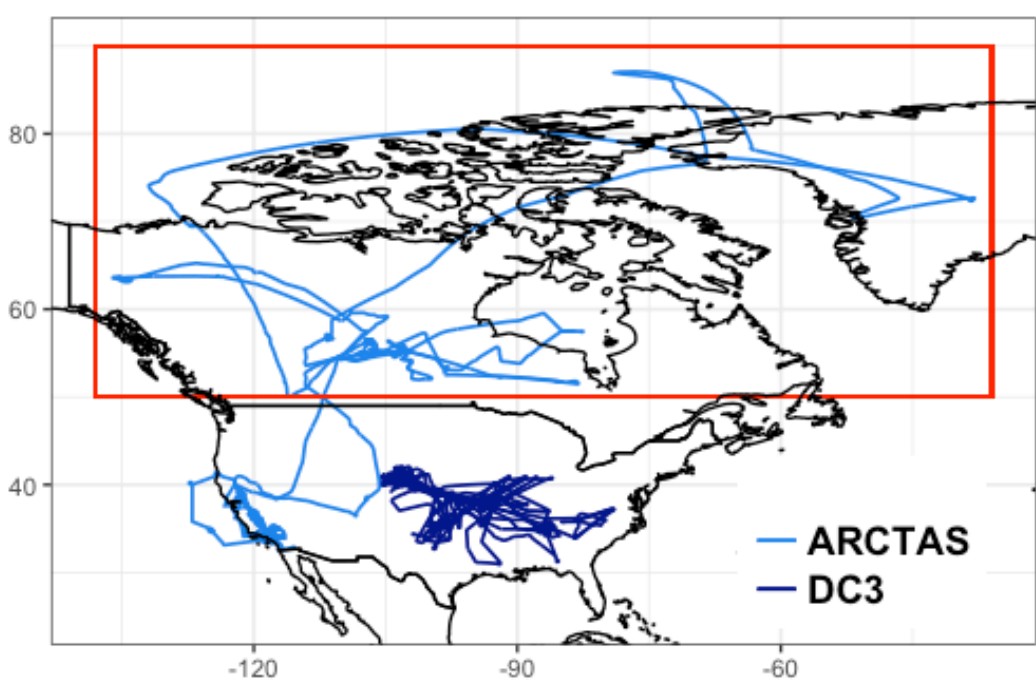


**Figure 1: Flight tracks of the ARCTAS and DC3 aircraft campaigns. The red box indicates the boreal region of the ARCTAS flights used here.**



**Figure 2: Boxplot summaries of each inventory's total annual emissions of BC, OC, and CO globally and for boreal North America and CONUS from 2004-2016. Diamonds indicate means. The horizontal bar is the median. The box shows the 25th to the 75th percentile, and the whiskers show 1.5 times the interquartile range. Points outside 1.5 times the interquartile range are shown as dots. GFED4s emissions are in red, FINN1.5 in orange, QFED2.4 in light blue, and GFAS1.2 in dark blue.**






**Figure 3: Emissions factors in g species/ kg DM (shown only for vegetated land) for each inventory over North America; BC shown on left, OC shown on right.**








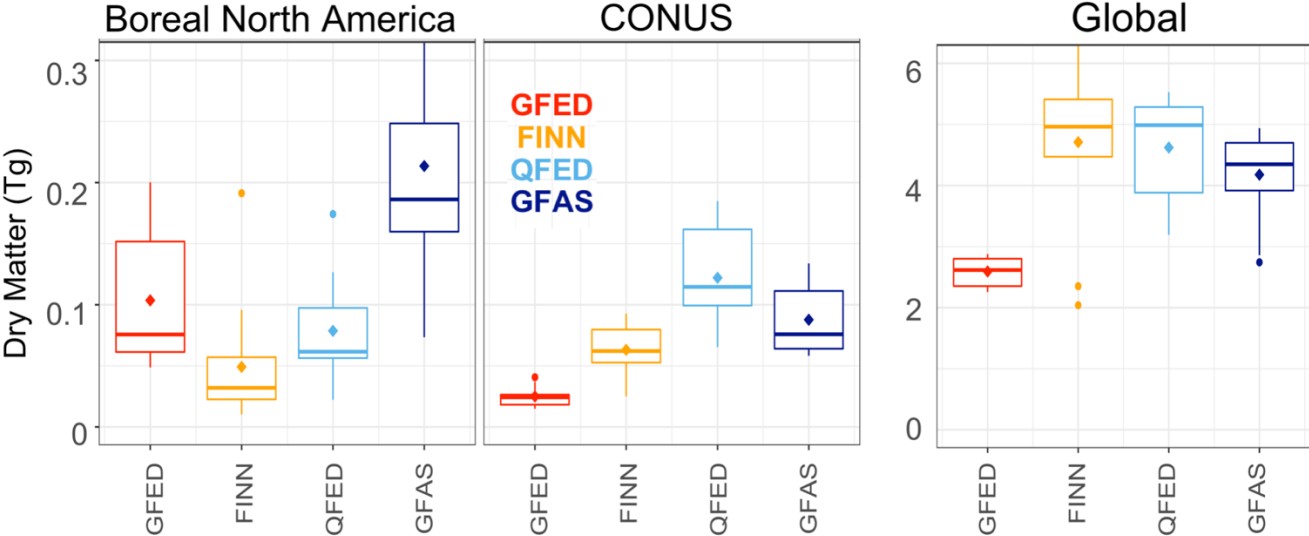


**Figure 4: Boxplot summary of each inventory's underlying total annual dry matter (DM) globally and for boreal North America and CONUS. The conventions of this boxplot are described in Fig. 2. GFED4s DM are in red, FINN1.5 in orange, QFED2.4 effective DM in light blue, and GFAS1.2 effective DM in dark blue.**






**Figure 5: Annual emissions scaled to GFED4s emissions factors from 2004-2016. The original inventory emissions from FINN1.5 (orange), QFED2.4 (light blue), and GFAS1.2 (dark blue) are shown as dashed lines while their annual values using GFED4s (red) emissions factors are shown as solid lines. 2012 is marked with a gray rectangle.**





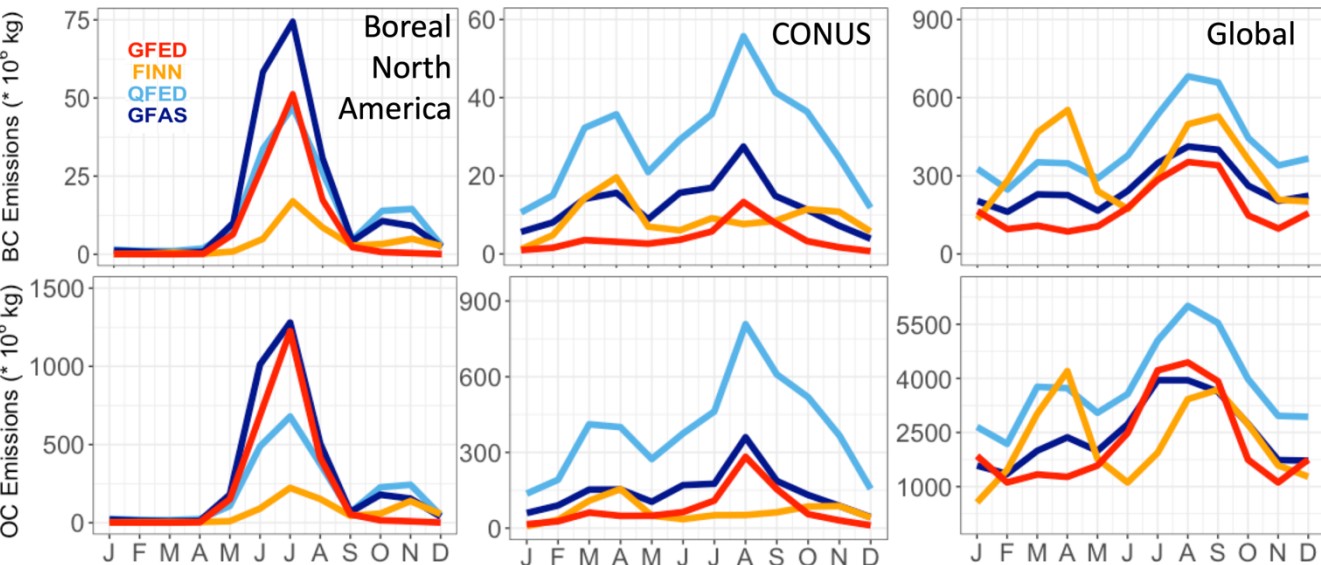

**Figure 6: Seasonal mean BC and OC emissions from 2004-2016 for boreal North America, CONUS, and the globe. GFED4s emissions are in red, FINN1.5 in orange, QFED2.4 in light blue, and GFAS1.2 in dark blue.**





**Figure 7: 2012 monthly comparison of simulated and observed median surface concentrations at IMPROVE sites in CONUS split between east and west at -104 degrees longitude. Observations in black are compared with concentrations simulated using GFED4s in red, FINN1.5 in orange, QFED2.4 in light blue, GFAS1.2 in dark blue, and a simulation with no biomass burning (noBB) in gray. Error bars show the 25th to 75th percentile range of observations. Note the different scales among panels.**

183

**Figure 8: Fire Season (May-September) 2012 mean surface BC and OA concentrations in CONUS with the model driven by each**
**inventory. Overlaid (circles) show mean observed surface concentrations at IMPROVE sites.**

186



**Figure 9: The median vertical profiles of BC and OA mass concentrations (shown in 0.5km bins) from the DC3 campaign. Observations (black) are compared with simulations using the four fire inventories– GFED4s (red), FINN1.5 (orange), QFED2.4 (light blue), and GFAS1.2 (dark blue) – and a simulation with no fire emissions (noBB) in gray. Error bars show the 25th – 75th percentile range of measurements averaged in each vertical bin. The number of observations in each bin is given on the right side of each panel. The left column shows total results for the campaign. The right column shows results filtered for the top 25th percentile of observed acetonitrile. Note the different scale between BC panels.**



**Figure 10:** The median vertical profiles of BC and OA mass concentrations (shown in 0.5km bins) from the boreal part of the ARCTAS campaign. Observations (black) are compared with simulations using the four fire inventories– GFED4s (red), FINN1.5 (orange), QFED2.4 (light blue), and GFAS1.2 (dark blue) – and a simulation with no fire emissions (noBB) in gray. Error bars show the $25^{th}$ – $75^{th}$ percentile range of measurements averaged in each vertical bin. The number of observations in each bin is given on the right side of each panel. The left column shows total results for the campaign. The right column shows results filtered for the top $25^{th}$ percentile of observed acetonitrile. Note the different scale among panels.






**Figure 11: The mean Northern Hemispheric fire season (May – September) 2012 and 2014 simulated AOD at 550nm sampled to and compared with daily MODIS-observed AOD from the Aqua satellite.**








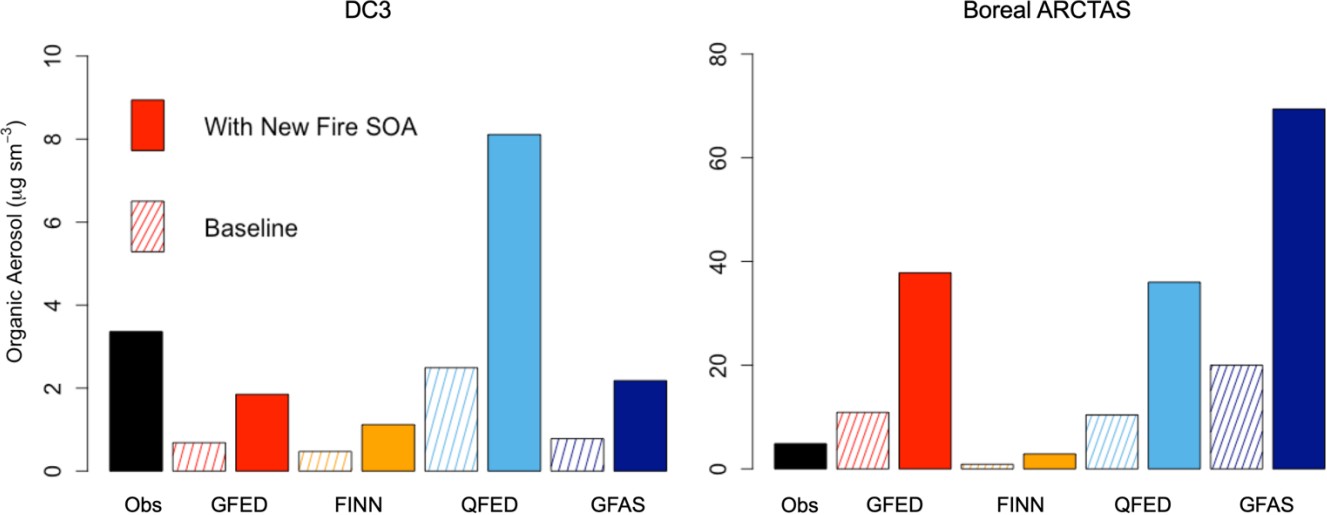


**Figure 12: Bar plots of mean OA mass concentrations from the DC3 (left panel) and boreal ARCTAS (right panel) campaigns. Observations (black) are compared with simulations using the four fire inventories– GFED4s (red), FINN1.5 (orange), QFED2.4 (light blue), and GFAS1.2 (dark blue). The hatched version of each inventory denotes OA mass concentrations using the baseline fire SOA scheme while the full color of each shows OA with the new SOA from fire parameterization.**



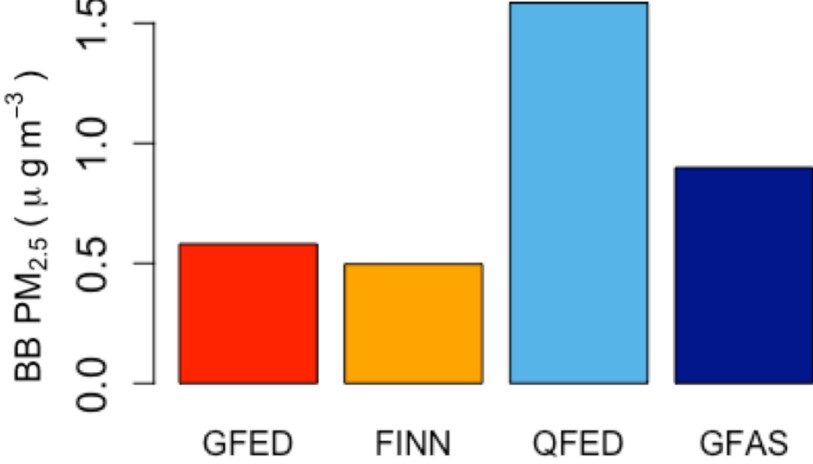


**Figure 13: Bar plots of the 2012 annual mean population-weighted fire PM2.5 exposure across the four inventories (GFED4s in red, FINN1.5 in orange, QFED2.4 in light blue, and GFAS1.2 in dark blue) across North America (Canada and CONUS only) at nested resolution. See Figure S6 for an analysis from 2012 – 2014 and for bar plots split out for Canada and the US at 2x2.5.**








**Figure 14: Top-of-atmosphere all-sky direct radiative effect of BB-only BC (top panel) and OA (bottom panel) averaged over 2012**
**in BONA, CONUS, and globally. GFED4s is shown in red, FINN1.5 orange, QFED2.4 light blue, and GFAS1.2 dark blue. (The size**
**of BC versus OA panels is not to scale).**

