# Peer review of "How emissions uncertainty influences the distribution and radiative impacts of smoke from fires in North America"

_Atmospheric Chemistry and Physics, 2019_

## Referee Comment (RC1) · Anonymous Referee #1 · 21 Nov 2019

The goal of this study presented in this paper is to quantify the uncertainties in the air quality and radiative impacts associated with BB aerosol emissions due to biomass (fuel) consumption and emission factors. The approach employed drives GEOS-Chem simulations with different BB emission inventories, two bottom-up (GFED4s and FINN) and two top-down (GFAS1.2 and QFED2.4). Results from the GEOS-Chem simulations are compared with observations of carbonaceous aerosols, black carbon (BC) and organic aerosol (OA), and satellite-derived aerosol optical depth (AOD) over North America. The authors find large differences in emissions and resultant air quality impacts across the four inventories explored. The authors conclude that differences between emission inventories in US, Alaska-Canada, and globally are driven by differ-

ences in fuel consumption, not emission factors.

The topic of this study is of great interest to the atmospheric chemistry and climate communities. The study was well designed and the authors are careful in their interpretation of the results. The manuscript is very well written and the presentation of methods and results is thorough, yet concise. I have only one significant issue with the paper and that is Section 6 (see comments below).

Specific Comments

L41: define AR5

L49-50: It should be noted that deep penetration of lungs and most acute health impacts are generally associated with fine PM (PM2.5) fraction of PM.

L78-79: This insertion of "climate forcing" seems incorrect. Even if one defines climate forcing as a perturbation of the Earth's energy balance, it need not be anthropogenic. Large volcanic eruptions result in climate forcing. And anthropogenic activities, e.g. land use and fire suppression (Andela et al., 2017), can reduce natural fire activity.

L169-170: Did the authors select the WRAP profile as opposed to Mu et al. (2011) since the focus was North America? Were there any complications/problems with using the WRAP cycle, which is intended to represent western US wildfires, for fires globally?

Mu et al. (2011) Daily and 3‐hourly variability in global fire emissions and consequences for atmospheric model predictions of carbon monoxide, JGR Atmospheres, 116, D24303

L247-256: The authors should better describe the challenge of FRP methods using MODIS data associated with the sparse temporal coverage. The most significant weakness in FRP based methods using observations from MODIS is the need to estimate FRP (for time integration to get FRE) between temporally sparse observations. Under cloud free conditions, in mid-latitudes Terra & Aqua provide 4 observations a day, maybe 6 depending on swath overlaps. FRP methods often require estimating FRP

between the Aqua over-pass ~13:30 LT and Terra ~22:30 LT over-pass, the period of peak fire activity in the western US and western Canada. (At high latitudes swath overlap increases and temporal coverage is much better).

L266-267: 0.16 ug/m3 several orders of magnitude lower than typical field BBOA concentrations? That suggests typical field BBOA around 2000 ug/m3. Is this typical? PM1 level of 2000 ug/m3 seems like a somewhat concentrated smoke plume. Please clarify.

L336-338: Liu et al. (2017) findings imply that the EFPM values used for western US wildfires may be higher than those used shown Fig 3. This should be clarified.

L343-353: Are the results for Boreal NA and CONUS similar if one uses EFOC to derive DM?

L383-384: It is likely that prescribed understory burning of forests in the southeast US are also a significant contributor to the CONUS springtime peak.

L395-398: "The southeastern US, in particular, 395 is of interest to the public health and policy communities because a prevalence of agricultural burning there, which dominates burned surface area (Nowell et al. 2018), may have a stronger impact on low altitude air quality in a relative sense than large wildfires that inject higher into the air."

While agricultural fires comprise a large share of fires and area burned in the southeast it does not dominate surface burned area. Nowell et al. (2018) reports that in Florida "silviculture fires consumed the most area (5.5 $\pm$ 6.7 $\times$ 105 ha/year), burning 50% more than agricultural fires."

More broadly across the southeast, the 2018 National Prescribed Fire Use Survey Report conducted by the National Association of State Foresters and the Coalition of Prescribed Fire Councils reported that 77% of prescribed fire acres burned in 2017 were forestry related compared to 23% agricultural. This breakdown is comparable to previous surveys released by the organizations in 2012 and 2015.

https://www.stateforesters.org/newsroom/nasf-coalition-of-prescribed-fire-councils-release-national-survey-on-prescribed-fire-use/

https://www.stateforesters.org/wp-content/uploads/2018/12/2018-Prescribed-Fire-Use-Survey-Report-1.pdf

L454-461: It would be helpful compare FiNN burned area versus GFED over CONUS. Maybe just add a sentence comparing average annual burned area. I suspect this would indicate a large difference in burned area, especially in the west, as the authors suggest. GFAS and QFED estimate/interpolate FRP between MODIS observations, essentially gap-filling for the large time periods without observations.

Section 6: While I find this analysis and interpretation valid, it leaves the impression BB smoke is not relevant wrt population exposure. I believe the 24 hour average PM2.5 is the metric that should be used for BB health impacts (35 ug/m3 per NAAQS). Day to week length exposures to wildfire smoke are associated with negative health impacts, see e.g. Liu et al. (2015), Fisk and Chan (2017), Moeltner et al (2013), Williamson et al. (2016). In the western US, days with high PM2.5 or days where PM2.5 exceeds the NAAQS standard, tend to associated with BB smoke (Liu et al., 2016; Brey et al., 2018; McClure and Jaffe, 2018). Section 6 should have focused on 24 hour average PM. I do not think it is necessary for the authors to so; however, I think it would improve the study and perhaps without significant extra effort. If the authors decide to not include an analysis based on 24 hour average PM, then they need to discuss smoke – health impact linkages associated with day – week(s) exposure.

Fisk and Chan (2017) Indoor Air, 27, 191–204

Liu et al. (2015) Environmental Research, 136, 120-132

Liu et al. (2016) Climatic Change, 138, 655-666

Moeltner et al. (2013) Journal of Environmental Economics and Management, 66, 476-496

McClure and Jaffe (2018), PNAS, 115, 7901-7906

Williamson et al. (2016) Environmental Research Letters, 11, 125009

Technical Comments

L29-32: Jumbled / missing text sentence L164: "OA" or "POA"?
* * *

---

## Referee Comment (RC2) · Anonymous Referee #3 · 19 Dec 2019

The study by Carter et al. uses a chemical transport model to compare four widely used fire emission inventories and assess the diversity in emitted and simulated quantities of biomass burning aerosol (BBA). To examine the performance of the model driven by the different fire emission inventories, model simulations are evaluated against in-situ and remote sensing observations. Implications of the diversity in fire emission estimates on air quality and aerosol radiative effects are also quantified and discussed. The paper is well written and structured, and the figures are well presented and clear. The concept of the study is certainly within the scope of ACP and I believe it will be very useful to both developers and users of these fire emissions datasets. I have a few minor general and specific comments (listed below) and I strongly recommend publication in ACP

once they have been addressed.

General comment

This study does a good job of exploring some uncertainties associated with quantifying fire emissions and the diversity between the different fire emissions datasets; and the subsequent impacts of these on simulated BBA. However, I would argue that without a full sensitivity analysis the study cannot fully quantify the magnitude and causes of uncertainty in simulated BBA. In reality, the sensitivity of simulated BBA to uncertainties in fire emissions is likely to be much larger than estimated in the paper because (as acknowledged by the authors e.g. in Section 3 paragraph 3) additional factors that are not considered may increase the estimated uncertainty range in fire emissions of BBA and secondly, because the sensitivity of simulated BBA to uncertain parameters is assessed one-at-a-time and interactions are not considered. There is some discussion in the paper in relation to the former e.g. L115-116 and L370, however it could be made clearer that the quantified "uncertainty" range is really the diversity range between emission datasets and that the full uncertainty range in fire emissions is yet to be quantified.

Specific comments

1. Abstract (P1, L20): "We aim to quantify the uncertainties associated with fire emissions..." related to the general comment above, I would say the study aims to explore the uncertainties rather than quantify them.

2. Abstract (P2, L37-39): "...sizeable range in BBA population-weighted exposure..." Could you state the time period you quantify the population-weighted exposure for (year and averaging period) and stress that it's exposure to BBA PM2.5.

3. Introduction (P2, L48-49): "Because of their small size..." Here it would be better to identify that it's aerosol in the size fraction below 2.5 $\mu$m that can penetrate deep into the lungs.

4. Introduction (paragraph 1): can you include some more references from the epidemiological literature for specific health impacts of BBA?

5. Introduction (P2-3, L56-68): Nice summary of papers on in-plume SOA production. Could be worth adding that several studies (some already cited) have suggested that the limited net changes in SOA mass could be explained by a balance between SOA formation and dilution and evaporation of POA mass (e.g. Jolleys et al., 2015; May et al., 2015; Zhou et al., 2017; Morgan et al., 2019... etc.).

6. Introduction (P3, L79-80): "The uncertainty in fire radiative impacts has not been assessed." This sentence could be written in a clearer way as there has been some previous assessments of the influence of biomass burning emission uncertainty on aerosol radiative forcing (e.g. Carslaw et al., 2016; Hamilton et al., 2018...). Perhaps just add "in detail" to the end of the sentence or something similar to: "The uncertainty in fire radiative impacts due to uncertainty in fire emissions has not been assessed in detail."

7. Section 2.1 (P6, L168-169): Can you specify whether fire emissions are averaged evenly across the PBL or if there is a gradient applied? What is the typical (or peak) model height of the daytime PBL over North America? What is the average model (vertical) resolution in the PBL over North America?

8. Sections 2.3 & 2.4 (general): Really nice descriptions of the observations and their uncertainties. However, these uncertainties are not referred to or taken into account in the results section (when the model is evaluated against these observations). I'm guessing that this is because the variability in observations and the model structural and emission uncertainties likely far outweigh measurement uncertainties, but this should be mentioned.

9. Section 3 (P10, L312): "variable in this inventory (i.e., more variability from 2004-2016 as evidenced by the taller boxplots)". I'm not sure the term "taller boxplots" is clear here. Do you refer to the larger range between 25th and 75th percentiles for

QFED? Can you give the range?

10. Section 3 (P10, L314-315): Are you referring to the global mean total annual emissions here?

11. Section 3 (paragraph 2): Can you add a line or two about any differences/similarities in the spatial pattern of emissions between the datasets (just for CONUS)?

12. Section 4 (general): Can you give some numbers to quantify the model skill in the text so that the different simulations can be quantitatively compared? Perhaps give temporal correlation values and/or model bias where appropriate.

13. Section 6 (general): To calculate fire PM2.5 are the BC and OC mass fractions summed for aerosol smaller or equal to 2.5 ïA▪m? Is there any contribution to PM2.5 from "primary" sulphate?

14. Section 6 (general): Do you see a range in exposure due to the differences in spatial patterns of the fire emissions? Are there any years that stick out?

15. Figure 8: It is very difficult to distinguish the colours of the overlapping circles (the black outlines obscure the colour inside the circle), particularly in the west. I suggest either showing an average in crowded regions or perhaps overlay the circles instead and just show the top colours.

16. Figure 11: It is difficult to assess the magnitude of the difference between the model and observations in this figure. I suggest including a figure showing some quantification of the difference e.g. showing the spatial distribution of the absolute difference or model bias? This figure could be put in the supplementary material.

References included in the review above:

Carslaw, K. S. et al. Large contribution of natural aerosols to uncertainty in indirect forcing. Nature 503, 67–71, 2013.

Hamilton, D.S., Hantson, S., Scott, C.E. et al. Reassessment of pre-industrial fire emissions strongly affects anthropogenic aerosol forcing. Nat Commun 9, 3182, doi:10.1038/s41467-018-05592-9, 2018.

Jolleys, M. D., Coe, H., McFiggans, G., Taylor, J. W., O'Shea, S. J., Le Breton, M., Bauguitte, S. J.-B., Moller, S., Di Carlo, P., Aruffo, E.,Palmer, P. I., Lee, J. D., Percival, C. J., and Gallagher, M. W.: Properties and evolution of biomass burning organic aerosol from Canadian boreal forest fires, Atmos. Chem. Phys., 15, 3077–3095, https://doi.org/10.5194/acp-15-3077-2015, 2015.

May, A. A., Lee, T., McMeeking, G. R., Akagi, S., Sullivan, A. P., Urbanski, S., Yokelson, R. J., and Kreidenweis, S. M.: Observations and analysis of organic aerosol evolution in some prescribed fire smoke plumes, Atmos. Chem. Phys., 15, 6323–6335,https://doi.org/10.5194/acp-15-6323-2015, 2015.

Morgan, W. T., Allan, J. D., Bauguitte, S., Darbyshire, E., Flynn, M. J., Lee, J., Liu, D., Johnson, B., Haywood, J., Longo, K. M., Artaxo, P. E., and Coe, H.: Transformation and aging of biomass burning carbonaceous aerosol over tropical South America from aircraft in-situ measurements during SAMBBA, Atmos. Chem. Phys. Discuss., https://doi.org/10.5194/acp-2019-157, in review, 2019.

Zhou, S., Collier, S., Jaffe, D. A., Briggs, N. L., Hee, J., Sedlacek III, A. J., Kleinman, L., Onasch, T. B., and Zhang, Q.: Regional influence of wildfires on aerosol chemistry in the western US and insights into atmospheric aging of biomass burning organic aerosol, Atmos. Chem. Phys., 17, 2477–2493, https://doi.org/10.5194/acp-17-2477-2017, 2017.

---

## Author Comment (AC1) · 15 Jan 2020

**Reviewer 1**
The goal of this study presented in this paper is to quantify the uncertainties in the air quality and radiative impacts associated with BB aerosol emissions due to biomass (fuel) consumption and emission factors. The approach employed drives GEOS-Chem simulations with different BB emission inventories, two bottom-up (GFED4s and FINN) and two top-down (GFAS1.2 and QFED2.4). Results from the GEOS-Chem simulations are compared with observations of carbonaceous aerosols, black carbon (BC) and organic aerosol (OA), and satellite-derived aerosol optical depth (AOD) over North America. The authors find large differences in emissions and resultant air quality impacts across the four inventories explored. The authors conclude that differences between emission inventories in US, Alaska-Canada, and globally are driven by differences in fuel consumption, not emission factors.

The topic of this study is of great interest to the atmospheric chemistry and climate communities. The study was well designed and the authors are careful in their interpretation of the results. The manuscript is very well written and the presentation of methods and results is thorough, yet concise. I have only one significant issue with the paper and that is Section 6 (see comments below).

We thank the referee for their comments and questions regarding our submitted manuscript. Below we have provided a list of the referee's specific comments and our responses in blue for each point.

Specific Comments
L41: define AR5

Done.

L49-50: It should be noted that deep penetration of lungs and most acute health impacts are generally associated with fine PM (PM2.5) fraction of PM.

Thank you for your comment. We have added language to this effect.

L78-79: This insertion of "climate forcing" seems incorrect. Even if one defines climate forcing as a perturbation of the Earth's energy balance, it need not be anthropogenic. Large volcanic eruptions result in climate forcing. And anthropogenic activities, e.g. land use and fire suppression (Andela et al., 2017), can reduce natural fire activity.

We agree that this our statement could be clarified on this point, and have added specific reference to the fact that all human activity (including ignition, suppression, and changing of fuel availability) impacts the DRF of fires.

L169-170: Did the authors select the WRAP profile as opposed to Mu et al. (2011) since the focus was North America? Were there any complications/problems with using the WRAP cycle, which is intended to represent western US wildfires, for fires globally? Mu et al. (2011) Daily and 3ˇARˇ hourly variability in global fire emissions and consequences for atmospheric model predictions of carbon monoxide, JGR Atmospheres, 116, D24303

Yes, we selected the WRAP diurnal scale factors because of our North America focus and their previous use in studies focused there (Kim et al. 2015; Saide et al. 2015). We did not verify whether this accurately represents the diurnal cycle in other regions of the world, but as the focus of our analysis (particularly with respect to observational comparisons) is on North America, we do not expect this to impact our results.

L247-256: The authors should better describe the challenge of FRP methods using MODIS data associated with the sparse temporal coverage. The most significant weakness in FRP based methods using observations from MODIS is the need to estimate FRP (for time integration to get FRE) between temporally sparse observations. Under cloud free conditions, in mid-latitudes Terra & Aqua provide 4 observations a day, maybe 6 depending on swath overlaps. FRP methods often require estimating FRP between the Aqua over-pass _13:30 LT and Terra _22:30 LT over-pass, the period of peak fire activity in the western US and western Canada. (At high latitudes swath overlap increases and temporal coverage is much better).

Thank you for this comment; we have added some additional discussion of the particular challenges associated with FRP-based approaches to the manuscript.

L266-267: 0.16 ug/m3 several orders of magnitude lower than typical field BBOA concentrations? That suggests typical field BBOA around 2000 ug/m3. Is this typical? PM1 level of 2000 ug/m3 seems like a somewhat concentrated smoke plume. Please clarify.

Thank you for pointing this out. Typical field BBOA measurements are often >= 10 ug m$^{-3}$, which is multiple orders of magnitude; however, our statement was vague. We have clarified in text.

L336-338: Liu et al. (2017) findings imply that the EFPM values used for western US wildfires may be higher than those used shown Fig 3. This should be clarified.

Thank you for this comment. We have added some text on this, as well as a reference to the new Andreae (2019) compilation that recently came out, to the manuscript.

L343-353: Are the results for Boreal NA and CONUS similar if one uses EFOC to derive DM?

Yes, in terms of the calculated DM consumed, the numbers are nearly identical when using either the OC or BC EF and emissions pairing.

L383-384: It is likely that prescribed understory burning of forests in the southeast US are also a significant contributor to the CONUS springtime peak.

Thank you – we have clarified in text.

L395-398: "The southeastern US, in particular, 395 is of interest to the public health and policy communities because a prevalence of agricultural burning there, which dominates burned surface area (Nowell et al. 2018), may have a stronger impact on low altitude air quality in a relative

sense than large wildfires that inject higher into the air." While agricultural fires comprise a large share of fires and area burned in the southeast it does not dominate surface burned area. Nowell et al. (2018) reports that in Florida "silviculture fires consumed the most area (5.5 _ 6.7 _ 105 ha/year), burning 50% more than agricultural fires." More broadly across the southeast, the 2018 National Prescribed Fire Use Survey Report conducted by the National Association of State Foresters and the Coalition of Prescribed Fire Councils reported that 77% of prescribed fire acres burned in 2017 were forestry related compared to 23% agricultural. This breakdown is comparable to previous surveys released by the organizations in 2012 and 2015.

https://www.stateforesters.org/newsroom/nasf-coalition-of-prescribed-fire-councilsrelease-national-survey-on-prescribed-fire-use/
https://www.stateforesters.org/wp-content/uploads/2018/12/2018-Prescribed-Fire-Use-Survey-Report-1.pdf

We thank the reviewer for this comment and have clarified that the SEUS is dominated by prescribed and agricultural fires in text.

L454-461: It would be helpful compare FiNN burned area versus GFED over CONUS. Maybe just add a sentence comparing average annual burned area. I suspect this would indicate a large difference in burned area, especially in the west, as the authors suggest. GFAS and QFED estimate/interpolate FRP between MODIS observations, essentially gap-filling for the large time periods without observations.

We thank the reviewer for this suggestion and have briefly looked into the difference between annual average burned area for GFED and FINN over CONUS. We actually find that FINN produces somewhat larger average annual burned area than GFED. This remains consistent with our point that it is both the relationship between fire activity and dry matter consumed and lower EFs used by FINN that contribute most to the lower values seen with FINN as shown in Figures 2-6.

Section 6: While I find this analysis and interpretation valid, it leaves the impression BB smoke is not relevant wrt population exposure. I believe the 24 hour average PM2.5 is the metric that should be used for BB health impacts (35 ug/m3 per NAAQS). Day to week length exposures to wildfire smoke are associated with negative health impacts, see e.g. Liu et al. (2015), Fisk and Chan (2017), Moeltner et al (2013), Williamson et al. (2016). In the western US, days with high PM2.5 or days where PM2.5 exceeds the NAAQS standard, tend to associated with BB smoke (Liu et al., 2016; Brey et al., 2018; McClure and Jaffe, 2018). Section 6 should have focused on 24 hour average PM. I do not think it is necessary for the authors to so; however, I think it would improve the study and perhaps without significant extra effort. If the authors decide to not include an analysis based on 24 hour average PM, then they need to discuss smoke – health impact linkages associated with day – week(s) exposure.

Fisk and Chan (2017) Indoor Air, 27, 191–204
Liu et al. (2015) Environmental Research, 136, 120-132
Liu et al. (2016) Climatic Change, 138, 655-666
Moeltner et al. (2013) Journal of Environmental Economics and Management, 66,

476-496
McClure and Jaffe (2018), PNAS, 115, 7901-7906
Williamson et al. (2016) Environmental Research Letters, 11, 125009

We thank the reviewer for this insightful comment and have added a discussion of 24-hour average BBA contribution to $PM_{2.5}$ across 2012 in CONUS and how it differs when using different emissions estimates. We have also added a figure in the supplement showing the range in the number of NAAQS exceedances per gridbox due to BB-only in 2012 when using each inventory.

Technical Comments
L29-32: Jumbled / missing text sentence L164: "OA" or "POA"?

Thank you for pointing this out. We have clarified in text that we are referring to POA.

---

## Author Comment (AC2)

**Reviewer 2**
The study by Carter et al. uses a chemical transport model to compare four widely used fire emission inventories and assess the diversity in emitted and simulated quantities of biomass burning aerosol (BBA). To examine the performance of the model driven by the different fire emission inventories, model simulations are evaluated against in-situ and remote sensing observations. Implications of the diversity in fire emission estimates on air quality and aerosol radiative effects are also quantified and discussed. The paper is well written and structured, and the figures are well presented and clear. The concept of the study is certainly within the scope of ACP and I believe it will be very useful to both developers and users of these fire emissions datasets. I have a few minor general and specific comments (listed below) and I strongly recommend publication in ACP once they have been addressed.

We thank the referee for their comments and questions regarding our submitted manuscript. Below we have provided a list of the referee's specific comments and our responses in blue for each point.

General comment
This study does a good job of exploring some uncertainties associated with quantifying fire emissions and the diversity between the different fire emissions datasets; and the subsequent impacts of these on simulated BBA. However, I would argue that without a full sensitivity analysis the study cannot fully quantify the magnitude and causes of uncertainty in simulated BBA. In reality, the sensitivity of simulated BBA to uncertainties in fire emissions is likely to be much larger than estimated in the paper because (as acknowledged by the authors e.g. in Section 3 paragraph 3) additional factors that are not considered may increase the estimated uncertainty range in fire emissions of BBA and secondly, because the sensitivity of simulated BBA to uncertain parameters is assessed one-at-a-time and interactions are not considered. There is some discussion in the paper in relation to the former e.g. L115-116 and L370, however it could be made clearer that the quantified "uncertainty" range is really the diversity range between emission datasets and that the full uncertainty range in fire emissions is yet to be quantified.

We appreciate this comment and very much agree that our work does not fully assess the full uncertainty range in fire emissions. In the introduction, lines 121-122 clarify that we refer to "the spread across these inventories as the "uncertainty" in emissions; however, we note that additional factors, not represented by any of these inventories, may increase the true uncertainty in the estimated emissions." To address the reviewer's comment, we have added text to reiterate this statement in the conclusions.

Specific comments

1. Abstract (P1, L20): "We aim to quantify the uncertainties associated with fire emissions..." related to the general comment above, I would say the study aims to explore the uncertainties rather than quantify them.

Thank you for this comment. We have changed quantify to explore.

2. Abstract (P2, L37-39): "...sizeable range in BBA population-weighted exposure..." Could you state the time period you quantify the population-weighted exposure for (year and averaging period) and stress that it's exposure to BBA PM2.5.

We have clarified in text that this is: "2012 annual BBA PM$_{2.5}$ population-weighted exposure over Canada and the contiguous United States."

3. Introduction (P2, L48-49): "Because of their small size..." Here it would be better to identify that it's aerosol in the size fraction below 2.5μm that can penetrate deep into the lungs.

We have clarified in text.

4. Introduction (paragraph 1): can you include some more references from the epidemiological literature for specific health impacts of BBA?

The Reid et al. 2016 paper addresses the specific health impacts of BBA, and we have added other references to that effect.

5. Introduction (P2-3, L56-68): Nice summary of papers on in-plume SOA production. Could be worth adding that several studies (some already cited) have suggested that the limited net changes in SOA mass could be explained by a balance between SOA formation and dilution and evaporation of POA mass (e.g. Jolleys et al., 2015; May et al., 2015; Zhou et al., 2017; Morgan et al., 2019...etc.).

We agree with the reviewer, and this is addressed by our statement: "the majority of field studies have reported no secondary aerosol formation (above dilution-corrected POA concentrations…)."

6. Introduction (P3, L79-80): "The uncertainty in fire radiative impacts has not been assessed." This sentence could be written in a clearer way as there has been some previous assessments of the influence of biomass burning emission uncertainty on aerosol radiative forcing (e.g. Carslaw et al., 2016; Hamilton et al., 2018...). Perhaps just add "in detail" to the end of the sentence or something similar to: "The uncertainty in fire radiative impacts due to uncertainty in fire emissions has not been assessed in detail."

We have added "in detail" as suggested.

7. Section 2.1 (P6, L168-169): Can you specify whether fire emissions are averaged evenly across the PBL or if there is a gradient applied? What is the typical (or peak) model height of the daytime PBL over North America? What is the average model(vertical) resolution in the PBL over North America?

Thank you very much for this question. Fire emissions are emitted from the surface and then mixed throughout the whole PBL, which we have clarified in text. We use the VDIFF PBL mixing scheme (described in Lin et al. (2008, AE) and Lin et al. (2010, AE)) as currently

implemented in GEOS-Chem where the PBL height is taken from the meteorological dataset (in this case, MERRA-2).

8. Sections 2.3 & 2.4 (general): Really nice descriptions of the observations and their uncertainties. However, these uncertainties are not referred to or taken into ac-count in the results section (when the model is evaluated against these observations). I'm guessing that this is because the variability in observations and the model structural and emission uncertainties likely far outweigh measurement uncertainties, but this should be mentioned.

We have now stated this explicitly in the text.

9. Section 3 (P10, L312): "variable in this inventory (i.e., more variability from 2004-2016 as evidenced by the taller boxplots)". I'm not sure the term "taller boxplots" is clear here. Do you refer to the larger range between 25th and 75th percentiles for QFED? Can you give the range?

We have clarified in text that we mean the wider range between the 25$^{th}$ and 75$^{th}$ percentiles. We chose not to include the range as it is different across species.

10. Section 3 (P10, L314-315): Are you referring to the global mean total annual emissions here?

Yes, we have clarified in text.

11. Section 3 (paragraph 2): Can you add a line or two about any differences/similarities in the spatial pattern of emissions between the datasets (just for CONUS)?

We have added a couple sentences describing the spatial patterns.

12. Section 4 (general): Can you give some numbers to quantify the model skill in the text so that the different simulations can be quantitatively compared? Perhaps give temporal correlation values and/or model bias where appropriate.

We have added R$^2$ values for the spatial plots (Figure 8).

13. Section 6 (general): To calculate fire PM2.5 are the BC and OC mass fractions summed for aerosol smaller or equal to 2.5 ï¿Am? Is there any contribution to PM2.5 from "primary" sulphate?

To calculate fire PM$_{2.5}$, we sum the BC and OA mass fractions for aerosol under 2.5 microns and have defined this in text. We do not include any contribution from sulphate.

14. Section 6 (general): Do you see a range in exposure due to the differences in spatial patterns of the fire emissions? Are there any years that stick out?

We do see a range in exposure due to differences in the spatial patterns of fire emissions. 2014 continues to be an outlier year for QFED2.4, and FINN is also larger in this year than in others –

leading to a smaller and a more significant population-weighted exposure, respectively, as a result.

15. Figure 8: It is very difficult to distinguish the colours of the overlapping circles (the black outlines obscure the colour inside the circle), particularly in the west. I suggest either showing an average in crowded regions or perhaps overlay the circles instead and just show the top colours.

We appreciate the reviewer's comment and have removed state lines from this figure to enhance legibility.

16. Figure 11: It is difficult to assess the magnitude of the difference between the model and observations in this figure. I suggest including a figure showing some quantification of the difference e.g. showing the spatial distribution of the absolute difference or model bias? This figure could be put in the supplementary material.

This plot is mainly for qualitative use, but your point is well-taken. We have added a plot of the spatial distribution of the model bias to the supplement.

References included in the review above:

Carslaw, K. S. et al. Large contribution of natural aerosols to uncertainty in indirect forcing. Nature 503, 67–71, 2013.

Hamilton, D.S., Hantson, S., Scott, C.E. et al. Reassessment of pre-industrial fireemissions strongly affects anthropogenic aerosol forcing. Nat Commun 9, 3182,doi:10.1038/s41467-018-05592-9, 2018.

Jolleys, M. D., Coe, H., McFiggans, G., Taylor, J. W., O'Shea, S. J., Le Breton, M.,Bauguitte, S. J.-B., Moller, S., Di Carlo, P., Aruffo, E.,Palmer, P. I., Lee, J. D., Per-cival, C. J., and Gallagher, M. W.: Properties and evolution of biomass burning or-ganic aerosol from Canadian boreal forest fires, Atmos. Chem. Phys., 15, 3077–3095,https://doi.org/10.5194/acp-15-3077-2015, 2015.

May, A. A., Lee, T., McMeeking, G. R., Akagi, S., Sullivan, A. P., Urbanski, S., Yokel-son, R. J., and Kreidenweis, S. M.: Observations and analysis of organic aerosolevolution in some prescribed fire smoke plumes, Atmos. Chem. Phys., 15, 6323–6335,https://doi.org/10.5194/acp-15-6323-2015, 2015.

Morgan, W. T., Allan, J. D., Bauguitte, S., Darbyshire, E., Flynn, M. J., Lee, J., Liu,D., Johnson, B., Haywood, J., Longo, K. M., Artaxo, P. E., and Coe, H.: Transforma-tion and aging of biomass burning carbonaceous aerosol over tropical South Americafrom aircraft in-situ measurements during SAMBBA, Atmos. Chem. Phys. Discuss.,https://doi.org/10.5194/acp-2019-157, in review, 2019.

Zhou, S., Collier, S., Jaffe, D. A., Briggs, N. L., Hee, J., Sedlacek III, A. J., Kleinman,L., Onasch, T. B., and Zhang, Q.: Regional influence of wildfires on aerosol chemistryin the western

US and insights into atmospheric aging of biomass burning organicaerosol, Atmos. Chem. Phys., 17, 2477–2493, https://doi.org/10.5194/acp-17-2477-2017, 2017.